# Stochastic Chebyshev Gradient Descent for Spectral Optimization

**Insu Han**[1]**, Haim Avron**[2] **and Jinwoo Shin**[1,3]
[1]School of Electrical Engineering, Korea Advanced Institute of Science and Technology
[2]Department of Applied Mathematics, Tel Aviv University
[3]AItrics
{insu.han,jinwoos}@kaist.ac.kr    haimav@post.tau.ac.il

## Abstract

A large class of machine learning techniques requires the solution of optimization problems involving spectral functions of parametric matrices, e.g. log-determinant and nuclear norm. Unfortunately, computing the gradient of a spectral function is generally of cubic complexity, as such gradient descent methods are rather expensive for optimizing objectives involving the spectral function. Thus, one naturally turns to stochastic gradient methods in hope that they will provide a way to reduce or altogether avoid the computation of full gradients. However, here a new challenge appears: there is no straightforward way to compute unbiased stochastic gradients for spectral functions. In this paper, we develop unbiased stochastic gradients for spectral-sums, an important subclass of spectral functions. Our unbiased stochastic gradients are based on combining randomized trace estimators with stochastic truncation of the Chebyshev expansions. A careful design of the truncation distribution allows us to offer distributions that are variance-optimal, which is crucial for fast and stable convergence of stochastic gradient methods. We further leverage our proposed stochastic gradients to devise stochastic methods for objective functions involving spectral-sums, and rigorously analyze their convergence rate. The utility of our methods is demonstrated in numerical experiments.

## 1 Introduction

A large class of machine learning techniques involves *spectral optimization* problems of the form,

$$\min_{\theta \in \mathcal{C}} F(A(\theta)) + g(\theta), \tag{1}$$

where $\mathcal{C}$ is some finite-dimensional parameter space, $A$ is a function that maps a parameter vector $\theta$ to a symmetric matrix $A(\theta)$, $F$ is a *spectral function* (i.e., a real-valued function on symmetric matrices that depends only on the eigenvalues of the input matrix), and $g : \mathcal{C} \to \mathbb{R}$. Examples include hyperparameter learning in Gaussian process regression with $F(X) = \log \det X$ [22], nuclear norm regularization with $F(X) = \mathtt{tr}\left(X^{1/2}\right)$ [20], phase retrieval with $F(X) = \mathtt{tr}\left(X\right)$ [8], and quantum state tomography with $F(X) = \mathtt{tr}\left(X \log X\right)$ [15]. In the aforementioned applications, the main difficulty in solving problems of the form (1) is in efficiently addressing the spectral component $F(A(\cdot))$. While explicit formulas for the gradients of spectral functions can be derived [17], it is typically computationally expensive. For example, for $F(X) = \log \det X$ and $A(\theta) \in \mathbb{R}^{d \times d}$, the exact computation of $\nabla_\theta F(A(\theta))$ can take as much as $O(d^3 k)$, where $k$ is the number of parameters in $\theta$. Therefore, it is desirable to avoid computing, or at the very least reduce the number of times we compute, the gradient of $F(A(\theta))$ exactly.

It is now well appreciated in the machine learning literature that the use of stochastic gradients is effective in alleviating costs associated with expensive exact gradient computations. Using cheap stochastic gradients, one can avoid computing full gradients altogether by using Stochastic Gradient Descent (SGD). The cost is, naturally, a reduced rate of convergence. Nevertheless, many machine learning applications require only mild suboptimality, in which case cheap iterations often outweigh the reduced convergence rate. When nearly optimal solutions are sought, more recent variance reduced methods (e.g. SVRG [14]) are effective in reducing the number of full gradient computations to $O(1)$. For non-convex objectives, the stochastic methods are even more attractive to use as they allow to avoid a bad local optimum. However, closed-form formulas for computing the full gradients of spectral functions do not lead to efficient stochastic gradients in a straightforward manner.

**Contribution.** In this paper, we propose stochastic methods for solving (1) when the spectral function $F$ is a *spectral-sum*. Formally, spectral-sums are spectral functions that can be expressed as $F(X) = \texttt{tr}\left(f(X)\right)$ where $f$ is a real-valued function that is lifted to the symmetric matrix domain by applying it to the eigenvalues. They constitute an important subclass of spectral functions, e.g., in all of the aforementioned applications of spectral optimization, the spectral function $F$ is a spectral-sum.

Our algorithms are based on recent *biased* estimators for spectral-sums that combine stochastic trace estimation with Chebyshev expansion [11]. The technique used to derive these estimators can also be used to derive stochastic estimators for the gradient of spectral-sums (e.g., see [7]), but the resulting estimator is biased. To address this issue, we propose an *unbiased* estimator for spectral-sums, and use it to derive unbiased stochastic gradients. Our unbiased estimator is based on randomly selecting the truncation degree in the Chebyshev expansion, i.e., the truncated polynomial degree is drawn under some distribution. We remark that similar ideas of sampling unbiased polynomials have been studied in the literature, but for different setups [4, 16, 28, 25], and none of which are suitable for use in our setup.

While deriving unbiased estimators is very useful for ensuring stable convergence of stochastic gradient methods, it is not sufficient: convergence rates of stochastic gradient descent methods depend on the variance of the stochastic gradients, and this can be rather large for naïve choices of degree distributions. Thus, our main contribution is in establishing a provably optimal degree distribution minimizing the estimators' variances with respect to the Chebyshev series. The proposed distribution gives order-of-magnitude smaller variances compared to other popular ones (Figure 1), which leads to improved convergence of the downstream optimization (Figure 2(c)).

We leverage our proposed unbiased estimators to design two stochastic gradient descent methods, one using the SGD framework and the other using the SVRG one. We rigorously analyze their convergence rates, showing sublinear and linear rate for SGD and SVRG, respectively. It is important to stress that our fast convergence results crucially depend on the proposed optimal degree distributions. Finally, we apply our algorithms to two machine learning tasks that involve spectral optimization: matrix completion and learning Gaussian processes. Our experimental results confirm that the proposed algorithms are significantly faster than other competitors under large-scale real-world instances. In particular, for learning Gaussian process under Szeged humid dataset, our generic method runs up to six times faster than the state-of-art method [7] specialized for the purpose.

## 2 Preliminaries

We denote the family of real symmetric matrices of dimension $d$ by $\mathcal{S}^{d \times d}$. For $A \in \mathcal{S}^{d \times d}$, we use $\|A\|_{\texttt{mv}}$ to denote the time-complexity of multiplying $A$ with a vector, i.e., $\|A\|_{\texttt{mv}} = O(d^2)$. For some structured matrices, e.g. low-rank, sparse or Toeplitz matrices, it is possible to have $\|A\|_{\texttt{mv}} = o(d^2)$.

### 2.1 Chebyshev expansion

Let $f : \mathbb{R} \to \mathbb{R}$ be an analytic function on $[a, b]$ for $a, b \in \mathbb{R}$. Then, the Chebyshev series of $f$ is given by

$$f(x) = \sum_{j=0}^{\infty} b_j T_j \left( \frac{2}{b-a} x - \frac{b+a}{b-a} \right), \quad b_j = \frac{2 - \mathbb{1}_{j=0}}{\pi} \int_{-1}^{1} \frac{f\left( \frac{b-a}{2} x + \frac{b+a}{2} \right) T_j(x)}{\sqrt{1 - x^2}} dx.$$

In the above, $\mathbb{1}_{j=0} = 1$ if $j = 0$ and $0$ otherwise and $T_j(x)$ is the Chebyshev polynomial (of the first kind) of degree $j$. An important property of the Chebyshev polynomials is the following recursive

formula: $T_{j+1}(x) = 2xT_j(x) - T_{j-1}(x)$, $T_1(x) = x$, $T_0(x) = 1$. The Chebyshev series can be used to approximate $f(x)$ via simply truncating the higher order terms, i.e., $f(x) \approx p_n(x) := \sum_{j=0}^{n} b_j T_j(\frac{2}{b-a}x - \frac{b+a}{b-a})$. We call $p_n(x)$ the *truncated Chebyhshev series* of degree $n$. For analytic functions, the approximation error (in the uniform norm) is known to decay exponentially [26]. Specifically, if $f$ is analytic with $\left|f(\frac{b-a}{2}z + \frac{b+a}{2})\right| \leq U$ for some $U > 0$ in the region bounded by the ellipse with foci $+1, -1$ and sum of major and minor semi-axis lengths equals to $\rho > 1$, then

$$|b_j| \leq \frac{2U}{\rho^j}, \quad \forall\, j \geq 0, \qquad \sup_{x \in [a,b]} |f(x) - p_n(x)| \leq \frac{4U}{(\rho - 1)\,\rho^n}. \tag{2}$$

## 2.2  Spectral-sums and their Chebyshev approximations

Given a matrix $A \in \mathcal{S}^{d \times d}$ and a function $f : \mathbb{R} \to \mathbb{R}$, the *spectral-sum* of $A$ with respect to $f$ is

$$\Sigma_f(A) := \texttt{tr}\,(f(A)) = \sum_{i=1}^{d} f(\lambda_i),$$

where $\texttt{tr}\,(\cdot)$ is the matrix trace and $\lambda_1, \lambda_2, \ldots, \lambda_d$ are the eigenvalues of $A$. Spectral-sums constitute an important subclass of spectral functions, and many applications of spectral optimization involve spectral-sums. This is fortunate since spectral-sums can be well approximated using Chebyshev approximations.

For a general $f$, one needs all eigenvalues to compute $\Sigma_f(A)$, while for some functions, simpler types of decomposition might suffice (e.g., $\log \det A = \Sigma_{\log}(A)$ can be computed using the Cholesky decomposition). Therefore, the general complexity of computing spectral-sums is $O(d^3)$, which is clearly not feasible when $d$ is very large, as is common in many machine learning applications. Hence, it is not surprising that recent literature proposed methods to approximate the large-scale spectral-sums, e.g., [11] recently suggested a fast randomized algorithm for approximating spectral-sums based on Chebyshev series and Monte-Carlo trace estimators (i.e., Hutchinson's method [13]):

$$\Sigma_f(A) = \texttt{tr}\,(f(A)) \approx \texttt{tr}\,(p_n(A)) = \mathbf{E_v}\left[\mathbf{v}^\top p_n(A)\mathbf{v}\right] \approx \frac{1}{M}\sum_{k=1}^{M} \mathbf{v}^{(k)\top}\left(\sum_{j=0}^{n} b_j \mathbf{w}_j^{(k)}\right) \tag{3}$$

where $\mathbf{w}_{j+1}^{(k)} = 2\left(\frac{2}{b-a}A - \frac{b+a}{b-a}I\right)\mathbf{w}_j^{(k)} - \mathbf{w}_{j-1}^{(k)}$, $\mathbf{w}_1^{(k)} = \left(\frac{2}{b-a}A - \frac{b+a}{b-a}I\right)\mathbf{v}$, $\mathbf{w}_0^{(k)} = \mathbf{v}^{(k)}$, and $\{\mathbf{v}^{(k)}\}_{k=1}^{M}$ are Rademacher random vectors, i.e., each coordinate of $\mathbf{v}^{(k)}$ is an i.i.d. random variable in $\{-1, 1\}$ with equal probability $1/2$ [13, 2, 24]. The approximation (3) can be computed using only matrix-vector multiplications, vector-vector inner-products and vector-vector additions $O(Mn)$ times each. Thus, the time-complexity becomes $O(Mn\|A\|_{\texttt{mv}} + Mnd) = O(Mn\|A\|_{\texttt{mv}})$. In particular, when $Mn \ll d$ and $\|A\|_{\texttt{mv}} = o(d^2)$, the cost can be significantly cheaper than $O(d^3)$ of exact computation. We further note that to apply the approximation (3), a bound on the eigenvalues is necessary. For an upper bound, one can use fast power methods [6]; this does not hurt the total algorithm complexity (see [10]). A lower bound can be encforced by substituting $A$ with The lower bound can typically be ensured $A + \varepsilon I$ for some small $\varepsilon > 0$. We use these techniques in our numerical experiments.

We remark that one may consider other polynomial approximation schemes, e.g. Taylor, but we focus on the Chebyshev approximations since they are nearly optimal in approximation among polynomial series [19]. Another recently suggested powerful technique is *stochastic Lanczos quadrature* [27], however it is not suitable for our needs (our bias removal technique is not applicable for it).

## 3  Stochastic Chebyshev gradients of spectral-sums

Our main goal is to develop scalable methods for solving the following optimization problem:

$$\min_{\theta \in \mathcal{C} \subseteq \mathbb{R}^{d'}} \Sigma_f(A(\theta)) + g(\theta), \tag{4}$$

where $\mathcal{C} \subseteq \mathbb{R}^{d'}$ is a non-empty, closed and convex domain, $A : \mathbb{R}^{d'} \to \mathcal{S}^{d \times d}$ is a function of parameter $\theta = [\theta_i] \in \mathbb{R}^{d'}$ and $g : \mathbb{R}^{d'} \to \mathbb{R}$ is some function whose derivative with respect to

any parameter $\theta$ is computationally easy to obtain. Gradient-descent type methods are natural candidates for tackling such problems. However, while it is usually possible to compute the gradient of $\Sigma_f(A(\theta))$, this is typically very expensive. Thus, we turn to stochastic methods, like (projected) SGD [3, 31] and SVRG [14, 30]. In order to apply stochastic methods, one needs unbiased estimators of the gradient. The goal of this section is to propose a computationally efficient method to generate unbiased stochastic gradients of small variance for $\Sigma_f(A(\theta))$.

## 3.1 Stochastic Chebyshev gradients

**Biased stochastic gradients.** We begin by observing that if $f$ is a polynomial itself or the Chebyshev approximation is exact, i.e., $f(x) = p_n(x) = \sum_{j=0}^{n} b_j T_j(\frac{2}{b-a} x - \frac{b+a}{b-a})$, we have

$$\frac{\partial}{\partial \theta_i} \Sigma_{p_n}(A) = \frac{\partial}{\partial \theta_i} \mathtt{tr}\left(p_n(A)\right) = \frac{\partial}{\partial \theta_i} \mathbf{E}_{\mathbf{v}}\left[\mathbf{v}^\top p_n(A) \mathbf{v}\right] = \mathbf{E}_{\mathbf{v}}\left[\frac{\partial}{\partial \theta_i} \mathbf{v}^\top p_n(A) \mathbf{v}\right]$$

$$\approx \frac{1}{M} \sum_{k=1}^{M} \frac{\partial}{\partial \theta_i} \mathbf{v}^{(k)\top} p_n(A) \mathbf{v}^{(k)} = \frac{1}{M} \sum_{k=1}^{M} \mathbf{v}^{(k)\top} \left(\sum_{j=0}^{n} b_j \frac{\partial \mathbf{w}_j^{(k)}}{\partial \theta_i}\right)^1, \qquad (5)$$

where $\{\mathbf{v}^{(k)}\}_{k=1}^{M}$ are i.i.d. Rademacher random vectors and $\partial \mathbf{w}_j^{(k)}/\partial \theta_i$ are given by the following recursive formula:

$$\frac{\partial \mathbf{w}_{j+1}^{(k)}}{\partial \theta_i} = \frac{4}{b-a} \frac{\partial A}{\partial \theta_i} \mathbf{w}_j^{(k)} + 2 \widetilde{A} \frac{\partial \mathbf{w}_j^{(k)}}{\partial \theta_i} - \frac{\partial \mathbf{w}_{j-1}^{(k)}}{\partial \theta_i}, \quad \frac{\partial \mathbf{w}_1^{(k)}}{\partial \theta_i} = \frac{2}{b-a} \frac{\partial A}{\partial \theta_i} \mathbf{v}^{(k)}, \quad \frac{\partial \mathbf{w}_0^{(k)}}{\partial \theta_i} = \mathbf{0}, \quad (6)$$

and $\widetilde{A} = \frac{2}{b-a} A - \frac{b+a}{b-a} I$. We note that in order to compute (6) only matrix-vector products with $A$ and $\partial A/\partial \theta_i$ are needed. Thus, stochastic gradients of spectral-sums involving polynomials of degree $n$ can be computed in $O(Mn(\|A\|_{\mathtt{mv}} \, d' + \sum_{i=1}^{d'} \|\frac{\partial A}{\partial \theta_i}\|_{\mathtt{mv}}))$. As we shall see in Section 5, the complexity can be further reduced in certain cases. The above estimator can be leveraged to approximate gradients for spectral-sums of analytic functions via the truncated Chebyshev series: $\nabla_\theta \Sigma_f(A(\theta)) \approx \nabla_\theta \Sigma_{p_n}(A(\theta))$. Indeed, [7] recently explored this in the context of Gaussian process kernel learning. However, if $f$ is not a polynomial, the truncated Chebyshev series $p_n$ is not equal to $f$, so the above estimator is biased, i.e. $\nabla_\theta \Sigma_f(A) \neq \mathbf{E}[\nabla_\theta \mathbf{v}^\top p_n(A) \mathbf{v}]$. The biased stochastic gradients might hurt iterative stochastic optimization as biased errors accumulate over iterations.

**Unbiased stochastic gradients.** The estimators (3) and (5) are biased since they approximate an analytic function $f$ via a polynomial $p_n$ of fixed degree. Unless $f$ is a polynomial itself, there exists an $x_0$ (usually uncountably many) for which $f(x_0) \neq p_n(x_0)$, so if $A$ has an eigenvalue at $x_0$ we have $\Sigma_f(A) \neq \Sigma_{p_n}(A)$. Thus, one cannot hope that the estimator (3), let alone the gradient estimator (5), to be unbiased for *all* matrices $A$. To avoid deterministic truncation errors, we simply randomize the degree, i.e., design some distribution $\mathcal{D}$ on polynomials such that for every $x$ we have $\mathbf{E}_{p \sim \mathcal{D}}[p(x)] = f(x)$. This guarantees $\mathbf{E}_{p \sim \mathcal{D}}[\mathtt{tr}\left(p(A)\right)] = \Sigma_f(A)$ from the linearity of expectation.

We propose to build such a distribution on polynomials by using truncated Chebyshev expansions where the truncation degree is stochastic. Let $\{q_i\}_{i=0}^{\infty} \subseteq [0,1]$ be a set of numbers such that $\sum_{i=0}^{\infty} q_i = 1$ and $\sum_{i=r}^{\infty} q_i > 0$ for all $r \geq 0$. We now define for $r = 0, 1, \dots$

$$\widehat{p}_r(x) \coloneqq \sum_{j=0}^{r} \frac{b_j}{1 - \sum_{i=0}^{j-1} q_i} T_j\left(\frac{2}{b-a} x - \frac{b+a}{b-a}\right). \qquad (7)$$

Note that $\widehat{p}_r(x)$ can be obtained from $p_r(x)$ by re-weighting each coefficient according to $\{q_i\}_{i=0}^{\infty}$. Next, let $n$ be a random variable taking non-negative integer values, and defined according to $\Pr(n = r) = q_r$. Under certain conditions on $\{q_i\}$, $\widehat{p}_n(\cdot)$ can be used to derive unbiased estimators of $\Sigma_f(A)$ and $\nabla_\theta \Sigma_f(A)$ as stated in the following lemma.

**Lemma 1** *Suppose that $f$ is an analytic function and $\widehat{p}_n$ is the randomized Chebyshev series of $f$ in (7). Assume that the entries of $A$ are differentiable for $\theta \in \mathcal{C}'$, where $\mathcal{C}'$ is an open set containing $\mathcal{C}$, and that for $a, b \in \mathbb{R}$ all the eigenvalues of $A(\theta)$ for $\theta \in \mathcal{C}'$ are in $[a, b]$. For any degree*

*distribution on non-negative integers* $\{q_i \in (0,1) : \sum_{i=0}^{\infty} q_i = 1, \sum_{r=i}^{\infty} q_r > 0, \forall i \geq 0\}$ *satisfying* $\lim_{n \to \infty} \sum_{i=n+1}^{\infty} q_i \widehat{p}_n(x) = 0$ *for all* $x \in [a, b]$, *it holds*

$$\mathbf{E}_{\mathbf{v},n} \left[ \mathbf{v}^\top \widehat{p}_n(A) \mathbf{v} \right] = \Sigma_f(A), \qquad \mathbf{E}_{\mathbf{v},n} \left[ \nabla_\theta \mathbf{v}^\top \widehat{p}_n(A) \mathbf{v} \right] = \nabla_\theta \Sigma_f(A). \qquad (8)$$

*where the expectations are taken over the joint distribution on random degree $n$ and Rademacher random vector $\mathbf{v}$ (other randomized probing vectors can be used as well).*

The proof of Lemma 1 is given in the supplementary material. We emphasize that (8) holds for any distribution $\{q_i\}_{i=0}^{\infty}$ on non-negative integers for which the conditions stated in Lemma 1 hold, e.g., geometric, Poisson or negative binomial distribution.

## 3.2 Main result: optimal unbiased Chebyshev gradients

It is a well-known fact that stochastic gradient methods converge faster when the gradients have smaller variances. The variance of our proposed unbiased estimators crucially depends on the choice of the degree distribution, i.e., $\{q_i\}_{i=0}^{\infty}$. In this section, we design a degree distribution that is variance-optimal in some formal sense. The variance of our proposed degree distribution decays exponentially with the expected degree, and this is crucial for for the convergence analysis (Section 4).

The degrees-of-freedoms in choosing $\{q_i\}_{i=0}^{\infty}$ is infinite, which poses a challenge for devising low-variance distributions. Our approach is based on the following simplified analytic approach studying the scalar function $f$ in such a way that one can naturally expect that the resulting distribution $\{q_i\}_{i=0}^{\infty}$ also provides low-variance for the matrix cases of (8). We begin by defining the variance of randomized Chebyshev expansion (7) via the Chebyshev weighted norm as

$$\text{Var}_C(\widehat{p}_n) := \mathbf{E}_n \left[ \|\widehat{p}_n - f\|_C^2 \right], \quad \text{where} \quad \|g\|_C^2 := \int_{-1}^{1} \frac{g(\frac{b-a}{2}x + \frac{b+a}{2})^2}{\sqrt{1 - x^2}} dx. \qquad (9)$$

The primary reason why we consider the above variance is because by utilizing the orthogonality of Chebyshev polynomials we can derive an analytic expression for it.

**Lemma 2** *Suppose* $\{b_j\}_{j=0}^{\infty}$ *are coefficients of the Chebyshev series for analytic function $f$ and $\widehat{p}_n$ is its randomized Chebyshev expansion* (7). *Then, it holds that* $\text{Var}_C(\widehat{p}_n) = \frac{\pi}{2} \sum_{j=1}^{\infty} b_j^2 \left( \frac{\sum_{i=0}^{j-1} q_i}{1 - \sum_{i=0}^{j-1} q_i} \right).$

The proof of Lemma 2 is given in the supplementary material. One can observe from this result that the variance reduces as we assign larger masses to to high degrees (due to exponentially decaying property of $b_j$ (2)). However, using large degrees increases the computational complexity of computing the estimators. Hence, we aim to design a good distribution given some target complexity, i.e., the expected polynomial degree $N$. Namely, the minimization of $\text{Var}_C(\widehat{p}_n)$ should be constrained by $\sum_{i=1}^{\infty} i q_i = N$ for some parameter $N \geq 0$.

However, minimizing $\text{Var}_C(\widehat{p}_n)$ subject to the aforementioned constraints might be generally intractable as the number of variables $\{q_i\}_{i=0}^{\infty}$ is infinite and the algebraic structure of $\{b_j\}_{j=0}^{\infty}$ is arbitrary. Hence, in order to derive an analytic or closed-form solution, we relax the optimization. In particular, we suggest the following optimization to minimize an upper bound of the variance by utilizing $|b_j| \leq 2U\rho^{-j}$ from (2) as follows:

$$\min_{\{q_i\}_{i=0}^{\infty}} \sum_{j=1}^{\infty} \rho^{-2j} \left( \frac{\sum_{i=0}^{j-1} q_i}{1 - \sum_{i=0}^{j-1} q_i} \right) \quad \text{subject to} \quad \sum_{i=1}^{\infty} i q_i = N, \sum_{i=0}^{\infty} q_i = 1 \text{ and } q_i \geq 0. \qquad (10)$$

Figure 1(d) empirically demonstrates that $b_j^2 \approx c\rho^{-2j}$ for constant $c > 0$ under $f(x) = \log x$, in which case the above relaxed optimization (10) is nearly tight. The next theorem establishes that (10) has a closed-form solution, despite having infinite degrees-of-freedom. The theorem is applicable when knowing a $\rho > 1$ and a bound $U$ such that the function $f$ is analytic with $\left| f\left( \frac{b-a}{2}z + \frac{b+a}{2} \right) \right| \leq U$ in the complex region bounded by the ellipse with foci $+1, -1$ and sum of major and minor semi-axis lengths is equal to $\rho > 1$.

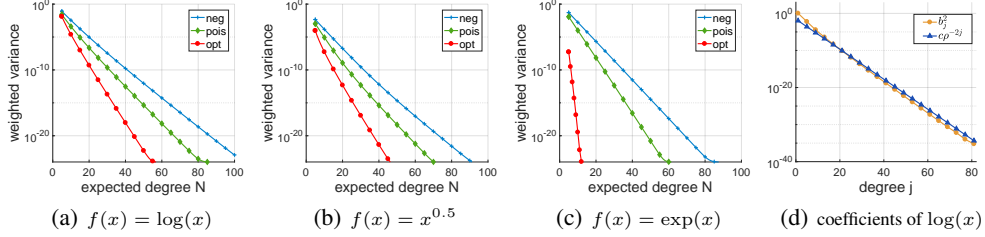

(a) $f(x) = \log(x)$    (b) $f(x) = x^{0.5}$    (c) $f(x) = \exp(x)$    (d) coefficients of $\log(x)$

Figure 1: Chebyshev weighted variance for three distinct distributions: negative binomial (neg), Poisson (pois) and the optimal distribution (11) (opt) with the same mean $N$ under (a) $\log x$, (b) $\sqrt{x}$ on $[0.05, 0.95]$ and (c) $\exp(x)$ on $[-1, 1]$, respectively. Observe that "opt" has the smallest variance among all distributions. (d) Comparison between $b_j^2$ and $c\rho^{-2j}$ for some constant $c > 0$ and $\log x$.

**Theorem 3** *Let $K = \max\{0, N - \left\lfloor \frac{\rho}{\rho-1} \right\rfloor\}$. The optimal solution $\{q_i^*\}_{i=0}^{\infty}$ of* (10) *is*

$$q_i^* = \begin{cases} 0 & \text{for } i < K \\ 1 - (N - K)(\rho - 1)\rho^{-1} & \text{for } i = K \\ (N - K)(\rho - 1)^2 \rho^{-i-1+N-K} & \text{for } i > K, \end{cases} \quad (11)$$

*and it satisfies the unbiasedness condition in Lemma 1, i.e., $\lim_{n\to\infty} \sum_{i=n+1}^{\infty} q_i^* \widehat{p}_n(x) = 0$.*

The proof of Theorem 3 is given in the supplementary material. Observe that a degree smaller than $K$ is never sampled under $\{q_i^*\}$, which means that the corresponding unbiased estimator (7) combines deterministic series of degree $K$ with randomized ones of higher degrees. Due to the geometric decay of $\{q_i^*\}$, large degrees will be sampled with exponentially small probability.

The optimality of the proposed distribution (11) (labeled opt) is illustrated by comparing it numerically to other distributions: negative binomial (labeled neg) and Poisson (labeled pois), on three analytic functions: $\log x$, $\sqrt{x}$ and $\exp(x)$. Figures 1(a) to 1(c) show the weighted variance (9) of these distributions where their means are commonly set from $N = 5$ to 100. Observe that the proposed distribution has order-of-magnitude smaller variance compared to other tested distributions.

## 4 Stochastic Chebyshev gradient descent algorithms

In this section, we leverage unbiased gradient estimators based on (8) in conjunction with our optimal degree distribution (11) to design computationally efficient methods for solving (4). In particular, we propose to randomly sample a degree $n$ from (11) and estimate the gradient via Monte-Carlo method:

$$\frac{\partial}{\partial \theta_i} \Sigma_f(A) = \mathbf{E}\left[\frac{\partial}{\partial \theta_i} \mathbf{v}^{\top} \widehat{p}_n(A) \mathbf{v}\right] \approx \frac{1}{M} \sum_{k=1}^{M} \mathbf{v}^{(k)\top}\left(\sum_{j=0}^{n} \frac{b_j}{1 - \sum_{i=0}^{j-1} q_i^*} \frac{\partial \mathbf{w}_j^{(k)}}{\partial \theta_i}\right) \quad (12)$$

where $\partial \mathbf{w}_j^{(k)} / \partial \theta_i$ can be computed using a Rademacher vector $\mathbf{v}^{(k)}$ and the recursive relation (6).

### 4.1 Stochastic Gradient Descent (SGD)

In this section, we consider the use of projected SGD in conjunction with (12) to numerically solve the optimization (4). In the following, we provide a pseudo-code description of our proposed algorithm.

---

**Algorithm 1** SGD for solving (4)

---

1: **Input:** number of iterations $T$, number of Rademacher vectors $M$, expected degree $N$ and $\theta^{(0)}$
2: **for** $t = 0$ to $T - 1$ **do**
3:    Draw $M$ Rademacher random vectors $\{\mathbf{v}^{(k)}\}_{k=1}^{M}$ and a random degree $n$ from (11) given $N$
4:    Compute $\psi^{(t)}$ from (12) at $\theta^{(t)}$ using $\{\mathbf{v}^{(k)}\}_{k=1}^{M}$ and $n$
5:    Obtain a proper step-size $\eta_t$
6:    $\theta^{(t+1)} \leftarrow \Pi_{\mathcal{C}}\left(\theta^{(t)} - \eta_t\left(\psi^{(t)} + \nabla g(\theta^{(t)})\right)\right)$, where $\Pi_{\mathcal{C}}(\cdot)$ is the projection mapping into $\mathcal{C}$
7: **end for**

---

In order to analyze the convergence rate, we assume that $(\mathcal{A}0)$ all eigenvalues of $A(\theta)$ for $\theta \in \mathcal{C}'$ are in the interval $[a, b]$ for some open $\mathcal{C}' \supseteq \mathcal{C}$, $(\mathcal{A}1)$ $\Sigma_f(A(\theta)) + g(\theta)$ is continuous and $\alpha$-strongly convex with respect to $\theta$ and $(\mathcal{A}2)$ $A(\theta)$ is $L_A$-Lipschitz for $\|\cdot\|_F$, $g(\theta)$ is $L_g$-Lipschitz and $\beta_g$-smooth. The formal definitions of the assumptions are in the supplementary material. These assumptions hold for many target applications, including the ones explored in Section 5. In particular, we note that assumption $(\mathcal{A}0)$ can be often satisfied with a careful choice of $\mathcal{C}$. It has been studied that (projected) SGD has a sublinear convergence rate for a smooth strongly-convex objective if the variance of gradient estimates is uniformly bounded [23, 21]. Motivated by this, we first derive the following upper bound on the variance of gradient estimators under the optimal degree distribution (11).

**Lemma 4** *Suppose that assumptions* $(\mathcal{A}0)$-$(\mathcal{A}2)$ *hold and* $A(\theta)$ *is* $L_{\mathrm{nuc}}$-*Lipschitz for* $\|\cdot\|_{\mathrm{nuc}}$. *Let* $\psi$ *be the gradient estimator* (12) *at* $\theta \in \mathcal{C}$ *using Rademacher vectors* $\{\mathbf{v}^{(k)}\}_{k=1}^{M}$ *and degree* $n$ *drawn from the optimal distribution* (11). *Then,* $\mathbf{E}_{\mathbf{v},n}[\|\psi\|_2^2] \leq \left(2L_A^2/M + d'L_{\mathrm{nuc}}^2\right)\left(C_1 + C_2N^4\rho^{-2N}\right)$ *where* $C_1, C_2 > 0$ *are some constants independent of* $M, N$.

The above lemma allows us to provide a sublinear convergence rate for Algorithm 1.

**Theorem 5** *Suppose that assumptions* $(\mathcal{A}0)$-$(\mathcal{A}2)$ *hold and* $A(\theta)$ *is* $L_{\mathrm{nuc}}$-*Lipschitz for* $\|\cdot\|_{\mathrm{nuc}}$. *If one chooses the step-size* $\eta_t = 1/\alpha t$, *then it holds that*

$$\mathbf{E}[\|\theta^{(T)} - \theta^*\|_2^2] \leq \frac{4}{\alpha^2 T}\max\left(L_g^2, \left(\frac{2L_A^2}{M} + d'L_{\mathrm{nuc}}^2\right)\left(C_1 + \frac{C_2N^4}{\rho^{2N}}\right)\right)$$

*where* $C_1, C_2 > 0$ *are constants independent of* $M, N$, *and* $\theta^* \in \mathcal{C}$ *is the global optimum of* (4).

The proofs of Lemma 4 and Theorem 5 are given in the supplementary material. Note that larger $M, N$ provide better convergence but they increase the computational complexity. The convergence is also faster with smaller $d'$, which is also evident in our experiments (see Section 5).

## 4.2 Stochastic Variance Reduced Gradient (SVRG)

In this section, we introduce a more advanced stochastic method using a further variance reduction technique, inspired by the stochastic variance reduced gradient method (SVRG) [14]. The full description of the proposed SVRG scheme for solving the optimization (4) is given below.

---

**Algorithm 2** SVRG for solving (4)

---

1: **Input:** number of inner/outer iterations $T, S$, number of Rademacher vectors $M$, expected degree $N$, step-size $\eta$ and initial parameter $\theta^{(0)} \in \mathcal{C}$
2: $\widetilde{\theta}^{(1)} \leftarrow \theta^{(0)}$
3: **for** $s = 1$ to $S$ **do**
4:     $\widetilde{\mu}^{(s)} \leftarrow \nabla\Sigma_f(A(\widetilde{\theta}^{(s)}))$ and $\theta^{(0)} \leftarrow \widetilde{\theta}^{(s)}$
5:     **for** $t = 0$ to $T - 1$ **do**
6:         Draw $M$ Rademacher random vectors $\{\mathbf{v}^{(k)}\}_{k=1}^{M}$ and a random degree $n$ from (11)
7:         Compute $\psi^{(t)}, \widetilde{\psi}^{(s)}$ from (12) at $\theta^{(t)}$ and $\widetilde{\theta}^{(s)}$, respectively using $\{\mathbf{v}^{(k)}\}_{k=1}^{M}$ and $n$
8:         $\theta^{(t+1)} \leftarrow \Pi_{\mathcal{C}}\left(\theta^{(t)} - \eta\left(\psi^{(t)} - \widetilde{\psi}^{(s)} + \widetilde{\mu}^{(s)} + \nabla g(\theta^{(t)})\right)\right)$
9:     **end for**
10:     $\widetilde{\theta}^{(s+1)} \leftarrow \frac{1}{T}\sum_{t=1}^{T}\theta^{(t)}$
11: **end for**

---

The main idea of SVRG is to subtract a mean-zero random variable to the original stochastic gradient estimator, where the randomness between them is shared. The SVRG algorithm was originally designed for optimizing finite-sum objectives, i.e., $\sum_i f_i(x)$, whose randomness is from the index $i$. On the other hand, the randomness in our case is from polynomial degrees and trace probing vectors for optimizing objectives of spectral-sums. This leads us to use the same randomness in $\{\mathbf{v}^{(k)}\}_{k=1}^{M}$ and $n$ for estimating both $\psi^{(t)}$ and $\widetilde{\psi}^{(s)}$ in line 7 of Algorithm 2. We remark that unlike SGD, Algorithm 2 requires the expensive computation of exact gradients every $T$ iterations. The next theorem establishes that if one sets $T$ correctly only $O(1)$ gradient computations are required (for a fixed suboptimality) since we have a linear convergence rate.

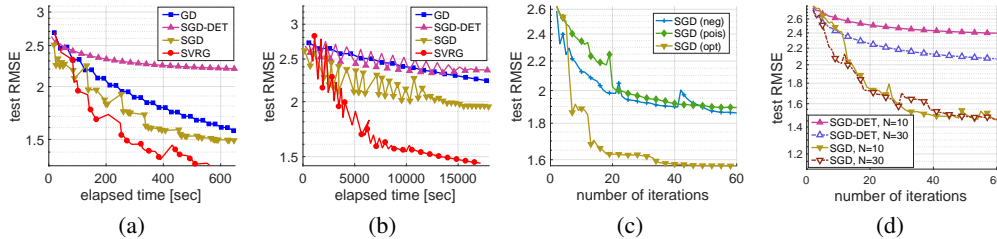

Figure 2: Matrix completion results under (a) MovieLens 1M and (b) MovieLens 10M. (c) Algorithm 1 (SGD) in MovieLens 1M under other distributions such as negative binomial (neg) and Poisson (pois). (d) SGD and SGD-DET under $N = 10, 30$.

**Theorem 6** *Suppose that assumptions* $(\mathcal{A}0)$-$(\mathcal{A}2)$ *hold and* $A(\theta)$ *is* $\beta_A$-*smooth for* $\|\cdot\|_F$. *Let* $\beta^2 = 2\beta_g^2 + \left(\frac{L_A^4 + \beta_A^2}{M} + L_A^4\right)\left(D_1 + \frac{D_2 N^8}{\rho^{2N}}\right)$ *for some constants* $D_1, D_2 > 0$ *independent of* $M, N$. *Choose* $\eta = \frac{\alpha}{7\beta^2}$ *and* $T \geq 25\beta^2/\alpha^2$. *Then, it holds that*

$$\mathbf{E}[\|\widetilde{\theta}^{(S)} - \theta^*\|_2^2] \leq r^S \mathbf{E}[\|\theta^{(0)} - \theta^*\|_2^2],$$

*where* $0 < r < 1$ *is some constant and* $\theta^* \in \mathcal{C}$ *is the global optimum of* (4).

The proof of the above theorem is given in the supplementary material, where we utilize the recent analysis of SVRG for the sum of smooth non-convex objectives [9, 1]. The key additional component in our analysis is to characterize $\beta > 0$ in terms of $M, N$ so that the unbiased gradient estimator (12) is $\beta$-smooth in expectation under the optimal degree distribution (11).

## 5 Applications

In this section, we apply the proposed methods to two machine learning tasks: matrix completion and learning Gaussian processes. These correspond to minimizing spectral-sums $\Sigma_f$ with $f(x) = x^{1/2}$ and $\log x$, respectively. We evaluate our methods under real-world datasets for both experiments.

### 5.1 Matrix completion

The goal is to recover a low-rank matrix $\theta \in [0, 5]^{d \times r}$ when a few of its entries are given. Since the rank function is neither differentiable nor convex, its relaxation such as Schatten-$p$ norm has been used in respective optimization formulations. In particular, we consider the smoothed nuclear norm (i.e., Schatten-1 norm) minimization [18, 20] that corresponds to

$$\min_{\theta \in [0,5]^{d \times r}} \mathtt{tr}(A^{1/2}) + \lambda \sum_{(i,j) \in \Omega} (\theta_{i,j} - R_{i,j})^2$$

where $A = \theta\theta^\top + \varepsilon I$, $R \in [0, 5]^{d \times r}$ is a given matrix with missing entries, $\Omega$ indicates the positions of known entries and $\lambda$ is a weight parameter and $\varepsilon > 0$ is a smoothing parameter. Observe that $\|A\|_{\mathtt{mv}} = \|\theta\|_{\mathtt{mv}} = O(dr)$, and the derivative estimation in this case can be amortized to compute using $O(dM(N^2 + Nr))$ operations. More details on this and our experimental settings are given in the supplementary material.

We use the MovieLens 1M and 10M datasets [12] (they correspond to $d = 3,706$ and $10,677$, respectively) and benchmark the gradient descent (GD), Algorithm 1 (SGD) and Algorithm 2 (SVRG). We also consider a variant of SGD using a deterministic polynomial degree, referred as SGD-DET, where it uses biased gradient estimators. We report the results for MovieLens 1M in Figure 2(a) and 10M in 2(b). For both datasets, SGD-DET performs badly due to its biased gradient estimators. On the other hand, SGD converges much faster and outperforms GD, where SGD for 10M converges much slower than that for 1M due to the larger dimension $d' = dr$ (see Theorem 5). Observe that SVRG is the fastest one, e.g., compared to GD, about 2 times faster to achieve RMSE 1.5 for MovieLens 1M and up to 6 times faster to achieve RMSE 1.8 for MovieLens 10M as shown in Figure 2(b). The gap between SVRG and GD is expected to increase for larger datasets. We also test SGD under other degree distributions: negative binomial (neg) and Poisson (pois) by choosing

parameters so that their means equal to $N = 15$. As reported in Figure 2(c), other distributions have relatively large variances so that they converge slower than the optimal distribution (opt). In Figure 2(d), we compare SGD-DET with SGD of the optimal distribution under the (mean) polynomial degrees $N = 10, 30$. Observe that a larger degree ($N = 30$) reduces the bias error in SGD-DET, while SGD achieves similar error regardless of the degree. The above results confirm that the unbiased gradient estimation and our degree distribution (11) are crucial for SGD.

## 5.2 Learning for Gaussian process regression

Next, we apply our method to hyperparameter learning for Gaussian process (GP) regression. Given training data $\left\{ \mathbf{x}_i \in \mathbb{R}^\ell \right\}_{i=1}^d$ with corresponding outputs $\mathbf{y} \in \mathbb{R}^d$, the goal of GP regression is to learn a hyperparameter $\theta$ for predicting the output of a new/test input. The hyperparameter $\theta$ constructs the kernel matrix $A(\theta) \in \mathcal{S}^{d \times d}$ of the training data $\{\mathbf{x}_i\}_{i=1}^d$ (see [22]). One can find a good hyperparameter by minimizing the negative log-marginal likelihood with respect to $\theta$:

$$\mathcal{L} := -\log p\left(\mathbf{y}|\{\mathbf{x}_i\}_{i=1}^d\right) = \frac{1}{2}\mathbf{y}^\top A(\theta)^{-1}\mathbf{y} + \frac{1}{2}\log\det A(\theta) + \frac{n}{2}\log 2\pi.$$

For handling large-scale datasets, [29] proposed the structured kernel interpolation framework assuming $\theta = [\theta_i] \in \mathbb{R}^3$ and

$$A(\theta) = WKW^\top + \theta_1^2 I, \quad K_{i,j} = \theta_2^2 \exp\left(\|\mathbf{x}_i - \mathbf{x}_j\|_2^2 / 2\theta_3^2\right),$$

where $W \in \mathbb{R}^{d \times r}$ is some sparse matrix and $K \in \mathbb{R}^{r \times r}$ is a dense kernel with $r \ll d$. Specifically, in [29], $r$ "inducing" points are selected and entries of $W$ are computed via interpolation with the inducing points. Under the framework, matrix-vector multiplications with $A$ can be performed even faster, requiring $\|A\|_{\mathtt{mv}} = \|W\|_{\mathtt{mv}} + \|K\|_{\mathtt{mv}} = O(d + r^2)$ operations. From $\|A\|_{\mathtt{mv}} = \|\frac{\partial A}{\partial \theta_i}\|_{\mathtt{mv}}$ and $d' = 3$, the complexity for computing gradient estimation (12) becomes $O(MN(d + r^2))$. If we choose $M, N, r = O(1)$, the complexity reduces to $O(d)$. The more detailed problem description and our experimental settings are given in the supplementary material.

We benchmark GP regression under natural sound dataset used in [29] and Szeged humid dataset [5] where they correspond to $d = 35,000$ and $16,930$, respectively. Recently, [7] utilized an approximation to derivatives of log-determinant based on stochastic Lanczos quadrature [27] (LANCZOS). We compare it with Algorithm 1 (SGD) which utilizes with unbiased gradient estimators while SVRG requires the exact gradient computation at least once which is intractable to run in these cases. As reported in Figure 3, SGD converges faster than LANCZOS for both datasets and it runs 2 times faster to achieve RMSE 0.0375 under sound dataset and under humid dataset LANCZOS can be often stuck at a local optimum, while SGD avoids it due to the use of unbiased gradient estimators.

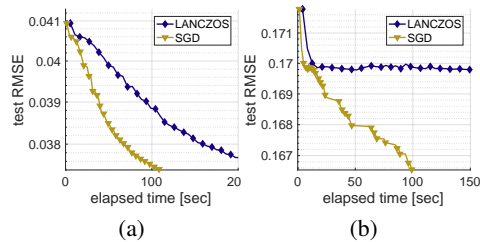

(a)                    (b)

Figure 3: Hyperparameter learning for Gaussian process in modeling (a) sound dataset and (b) Szeged humid dataset comparing SGD to stochastic Lanczos quadrature (LANCZOS).

## 6 Conclusion

We proposed an optimal variance unbiased estimator for spectral-sums and their gradients. We applied our estimator in the SGD and SVRG frameworks, and analyzed convergence. The proposed optimal degree distribution is a crucial component of the analysis. We believe that the proposed stochastic methods are of broader interest in many machine learning tasks involving spectral-sums.

## Acknowledgement

This work was supported by the National Research Foundation of Korea(NRF) grant funded by the Korea government(MSIT) (2018R1A5A1059921). Haim Avron acknowledges the support of the Israel Science Foundation (grant no. 1272/17).

## Footnotes

[1]We assume that all partial derivatives $\partial A_{j,k}/\partial \theta_i$ for $j, k = 1, \dots, d$, $i = 1, \dots, d'$ exist and are continuous.

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
