[Supplementary Material]

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

\|_{\texttt{mv}} = \|W\|_{\texttt{mv}} + \|K\|_{\texttt{mv}} = O(d + r^2)$ operations. From $\|A\|_{\texttt{mv}} = \|\frac{\partial A}{\partial \theta_i}\|_{\texttt{mv}}$ and $d' = 3$, the complexity for computing gradient estimation (12) becomes $O(MN(d + r^2))$. If we choose $M, N, r = O(1)$, the complexity reduces to $O(d)$. The more detailed problem description and our experimental settings are given in the supplementary material.

We benchmark GP regression under natural sound dataset used in [32] and Szeged humid dataset [6] where they correspond to $d = 35,000$ and $16,930$, respectively. Recently, [8] utilized an approximation to derivatives of log-determinant based on stochastic Lanczos quadrature [30] (LANCZOS). We compare it with Algorithm 1 (SGD) which utilizes with unbiased gradient estimators while SVRG requires the exact gradient computation at least once which is intractable to run in these cases. As reported in Figure 3, SGD converges faster than LANCZOS for both datasets and it runs 2 times faster to achieve RMSE 0.0375 under sound dataset and under humid dataset LANCZOS can be often stuck at a local optimum, while SGD avoids it due to the use of unbiased gradient estimators.

Figure 3: Hyperparameter learning for Gaussian process in modeling (a) sound dataset and (b) Szeged humid dataset comparing SGD to stochastic Lanczos quadrature (LANCZOS).

## 6 Conclusion

We proposed an optimal variance unbiased estimator for spectral-sums and their gradients. We applied our estimator in the SGD and SVRG frameworks, and analyzed convergence. The proposed optimal degree distribution is a crucial component of the analysis. We believe that the proposed stochastic methods are of broader interest in many machine learning tasks involving spectral-sums.

## Acknowledgement

This work was supported by the National Research Foundation of Korea(NRF) grant funded by the Korea government(MSIT) (2018R1A5A1059921). Haim Avron acknowledges the support of the Israel Science Foundation (grant no. 1272/17).

## Footnotes

[1] We assume that all partial derivatives $\partial A_{j,k}/\partial \theta_i$ for $j, k = 1, \dots, d, i = 1, \dots, d'$ exist and are continuous.

[2]If $k = -1$, it is equivalent to the minimum from Cauchy-Schwarz inequality.

[3] Indeed, $\mathbf{y}_j = U_j(A) \mathbf{v}$ for $j \geq 1$, where $U_j(x)$ is the $j$-th Chebyshev polynomial of the second-kind.

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

# Stochastic Chebyshev Gradient Descent for Spectral Optimization: Supplementary material

## A  Details of experiments

### A.1  Matrix completion

For matrix completion, the problem can be expressed via the convex smoothed nuclear norm minimization as

$$\min_{\theta \in [0,5]^{d \times r}} \mathtt{tr}(A^{1/2}) + \lambda \sum_{(i,j) \in \Omega} (\theta_{i,j} - R_{i,j})^2, \tag{13}$$

where $A = \theta\theta^\top + \varepsilon I$, $R \in [0,5]^{d \times r}$ is a given matrix with missing entries, $\Omega$ indicates the positions of known entries and $\lambda$ is a weight parameter and $\varepsilon > 0$ is a smoothing parameter. In this case, the gradient estimator (12) can be amortized as

$$\nabla_\theta \mathtt{tr}(A^{p/2}) \approx \frac{2}{M} \sum_{k=1}^{M} \sum_{i=0}^{n-1} (2 - \mathbb{1}_{i=0}) \mathbf{w}_i^{(k)} \left( \sum_{j=i}^{n-1} \frac{b_{j+1}}{1 - \sum_{\ell=0}^{j} q_\ell^*} \mathbf{y}_{j-i}^{(k)} \right)^\top \theta \tag{14}$$

where

$$\mathbf{w}_{j+1}^{(k)} = 2\mathbf{w}_j^{(k)} - \mathbf{w}_{j-1}^{(k)}, \quad \mathbf{w}_1^{(k)} = \widetilde{A}\mathbf{v}, \quad \mathbf{w}_0^{(k)} = \mathbf{v}^{(k)},$$
$$\mathbf{y}_{j+1}^{(k)} = 2\mathbf{w}_{j+1}^{(k)} + \mathbf{y}_{j-1}^{(k)}, \quad \mathbf{y}_1^{(k)} = 2\widetilde{A}\mathbf{v}^{(k)}, \quad \mathbf{y}_0^{(k)} = \mathbf{v}^{(k)}$$

and $\widetilde{A} = \left( \frac{2}{b-a}A - \frac{b+a}{b-a}I \right)$ for the lower/upper bound on $A$'s eigenvalues $a, b \in \mathbb{R}^+$. This comes from the following lemma, whose proof is in Section B.5.

**Lemma 7** *Suppose $f$ is an analytic function and $p_n(x) := \sum_{j=0}^{n} b_j T_j(x)$ is its truncated Chebyshev series of degree $n \geq 1$ for $x \in [-1, 1]$. Let $A = \theta\theta^\top + \varepsilon I$ for $\theta \in \mathbb{R}^{d \times r}, \varepsilon > 0$ such that all eigenvalues of $A$ are in $[-1, 1]$. Then, for any $\mathbf{v} \in \mathbb{R}^d$, it holds that*

$$\nabla_\theta \mathbf{v}^\top p_n(A)\mathbf{v} = 2 \sum_{i=0}^{n-1} (2 - \mathbb{1}_{i=0}) \mathbf{w}_i \left( \sum_{j=i}^{n-1} b_{j+1}\mathbf{y}_{j-i} \right)^\top \theta,$$

*where $\mathbf{w}_{j+1} = 2A\mathbf{w}_j - \mathbf{w}_{j-1}, \mathbf{w}_1 = A\mathbf{v}, \mathbf{w}_0 = \mathbf{v}$ and $\mathbf{y}_{j+1} = 2\mathbf{w}_{j+1} + \mathbf{y}_{j-1}, \mathbf{y}_1 = 2A\mathbf{v}, \mathbf{y}_0 = \mathbf{v}$.*

Observe that $\|A\|_{\mathtt{mv}} = \|\theta\|_{\mathtt{mv}} = O(dr)$, and the computation for (14) can be amortized using $O(M(n^2 d + ndr))$ operations. For $M, n, r = O(1)$, the complexity reduces to $O(d)$.

After update the parameter $\theta$ in a direction of gradient estimator, we project $\theta$ onto $[0,5]^{d \times r}$, that is,

$$\Pi_{\mathcal{C}}(\theta_{i,j}) = \begin{cases} \theta_{i,j}, & \text{if } \theta_{i,j} \in [0,5], \\ 0, & \text{if } \theta_{i,j} < 0, \\ 5, & \text{otherwise.} \end{cases}$$

In addition, after performing all gradient updates, we finally apply low-rank approximation using truncated SVD with rank 10 once and measure the test root mean square error (RMSE).

**Setup.** We use matrix $R$ from MovieLens 1M (about $10^6$ integer ratings from 1 to 5 from $6,040$ users on $3,706$ movies) and 10M (about $10^7$ ratings from 0.5 to 5 with intervals 0.5 from $10,677$

users on $71,567$ movies) datasets [13].We randomly select $90\%$ of each dataset for training and use the rest for testing. We choose the (mean) polynomial degree $N = 15$ and the number of trace random vectors $M = 100$ for SVRG and $M = 200$ for SGD-DET, SGD, respectively, for comparable complexity at each gradient update. Especially, for SVRG, we choose $T = 100$.We decrease step-sizes exponentially with ratio $0.97$ over the iterations.

## A.2 Gaussian process (GP) regression

Given training data $\left\{ \mathbf{x}_i \in \mathbb{R}^\ell \right\}_{i=1}^d$ with corresponding outputs $\mathbf{y} \in \mathbb{R}^d$, the goal of GP regression is to learn a hyperparameter $\theta$ for predicting the output of a new/test input. GP defines a distribution over functions, which follow multivariate Gaussian distribution with mean function $\mu_\theta : \mathbb{R}^\ell \to \mathbb{R}$ and covariance (i.e., kernel) function $a_\theta : \mathbb{R}^\ell \times \mathbb{R}^\ell \to \mathbb{R}$. To this end, we set the kernel matrix $A = A(\theta) \in \mathcal{S}^{d \times d}$ of $\{\mathbf{x}_i\}_{i=1}^d$ such that $A_{i,j} = a_\theta(\mathbf{x}_i, \mathbf{x}_j)$ and the mean function to be zero. One can find a good hyperparameter by minimizing the negative log-marginal likelihood with respect to $\theta$:

$$\mathcal{L} := -\log p \left( \mathbf{y} | \{\mathbf{x}_i\}_{i=1}^d \right) = \frac{1}{2}\mathbf{y}^\top A^{-1}\mathbf{y} + \frac{1}{2}\log \det A + \frac{n}{2}\log 2\pi, \qquad (15)$$

and predict $y = \mathbf{a}^\top A^{-1}\mathbf{y}$ where $\mathbf{a}_i = a_\theta(\mathbf{x}_i, \mathbf{x})$ (see [23]). Gradient-based methods can be used for optimizing (15) using its partial derivatives:

$$\frac{\partial \mathcal{L}}{\partial \theta_i} = -\frac{1}{2}\left( \mathbf{y}^\top \frac{\partial A}{\partial \theta_i} \right) A^{-1} \left( \frac{\partial A}{\partial \theta_i} \mathbf{y} \right) - \frac{1}{2}\frac{\partial \log \det A}{\partial \theta_i}.$$

Observe that the first term can be computed by an efficient linear solver, e.g., conjugate gradient descents [27], while the second term is computationally expensive for large $d$. Hence, one can use our proposed gradient estimator (12) for $\Sigma_f(A)$ with $f(x) = \log x$.

For handling large-scale datasets, [32] proposed the structured kernel interpolation framework assuming $\theta = [\theta_i] \in \mathbb{R}^3$ and

$$A(\theta) = WKW^\top + \theta_1^2 I, K_{i,j} = \theta_2^2 \exp \left( \|\mathbf{x}_i - \mathbf{x}_j\|_2^2 / 2\theta_3^2 \right),$$

where $W \in \mathbb{R}^{d \times r}$ is some sparse matrix and $K \in \mathbb{R}^{r \times r}$ is a dense kernel with $r \ll d$. Specifically, the authors select $r$ "inducing" points and compute entries of $W$ via interpolation with the inducing points. Under the framework, matrix-vector multiplications with $A$ can be performed even faster, requiring $\|A\|_{\mathtt{mv}} = \|W\|_{\mathtt{mv}} + \|K\|_{\mathtt{mv}} = O(d + r^2)$ operations. From $\|A\|_{\mathtt{mv}} = \|\frac{\partial A}{\partial \theta_i}\|_{\mathtt{mv}}$ and $d' = 3$, the complexity for computing gradient estimation (12) becomes $O(MN(d + r^2))$. If we choose $M, N, r = O(1)$, the complexity reduces to $O(d)$.

**Setup.** We benchmark GP regression under natural sound dataset used in [32, 8] and Szeged humid data [6]. We randomly choose $35,000$ points for training and $691$ for testing in sound dataset and choose $16,930$ points for training and $614$ points for test in Szeged 2015-2016 humid dataset. We set the polynomial degree $N = 15$ and $M = 30$ trace vectors for all algorithms. We also select $r = 3000$ induced points for kernel interpolation. Since GP regression is non-convex problem, the gradient descent methods are sensitive to the initial point. We select a good initial point using random grid search. We observe that our algorithm (SGD) utilizing unbiased gradient estimator performs well for any initial point. On the other hand, since LANCZOS is type of biased gradient descent methods, it is often stuck on a bad local optimum.

# B Proof of theorems

## B.1 Smoothness and strong convexity of matrix functions

We first provide the formal definitions of the assumptions in Section 4. Let $\mathcal{C} \subseteq \mathbb{R}^{d'}$ be a non-empty, closed convex domain and $h : \mathbb{R}^{d'} \to \mathbb{R}$ be a continuously differentiable function.

**Definition 1** *A function $h$ is L-Lipschitz continuous (or L-Lipschitz) on $\mathcal{C}$ if for all $\theta, \theta' \in \mathcal{C}$, there exists a constant $L > 0$ such that*

$$|h(\theta) - h(\theta')| \le L \|\theta - \theta'\|_2 .$$

**Definition 2** *A function $h$ is $\beta$-smooth on $\mathcal{C}$ if its gradient is $\beta$-Lipschitz such that*

$$\|\nabla h(\theta) - \nabla h(\theta')\|_2 \leq \beta \|\theta - \theta'\|_2.$$

**Definition 3** *A function $h$ is $\alpha$-strongly convex on $\mathcal{C}$ if for all $\theta, \theta' \in \mathcal{C}$, there exists a constant $\alpha > 0$ such that*

$$\langle \nabla h(\theta) - \nabla h(\theta'), \theta - \theta' \rangle \geq \alpha \|\theta - \theta'\|_2^2.$$

The above definition can be extended to functions map into matrix space. For example, suppose $A : \mathbb{R}^{d'} \to \mathbb{R}^{d \times d}$ is a function of $\theta \in \mathcal{C}$ and assume that all $\partial A_{j,k} / \partial \theta_i$ 's exist and are continuous.

**Definition 4** *A function $A(\theta)$ is $L_A$-Lipschitz with respect to $\|\cdot\|_F$ if for all $\theta, \theta' \in \mathcal{C}$, there exists a constant $L_A > 0$ such that*

$$\|A(\theta) - A(\theta')\|_F \leq L_A \|\theta - \theta'\|_2.$$

*Similarly, $A(\theta)$ is $L_{\mathrm{nuc}}$-Lipschitz with respect to $\|\cdot\|_{\mathrm{nuc}}$ (matrix nuclear norm) there exists a constant $L_{\mathrm{nuc}} > 0$ such that*

$$\|A(\theta) - A(\theta')\|_{\mathrm{nuc}} \leq L_{\mathrm{nuc}} \|\theta - \theta'\|_2.$$

**Definition 5** *Let $A : \mathbb{R}' \to \mathcal{S}^{d \times d}$ be a continuously differentiable function of $\theta \in \mathcal{C}$. If $A(\theta)$ is $\beta_A$-smooth if for all $\theta, \theta' \in \mathcal{C}$, there exists a constant $\beta_A > 0$ such that*

$$\left\| \frac{\partial A(\theta)}{\partial \theta} - \frac{\partial A(\theta')}{\partial \theta} \right\|_F \leq \beta_A \|\theta - \theta'\|_2.$$

## B.2  Proof of Theorem 3 : optimal degree distribution

By adding $\sum_{j=1}^{\infty} \rho^{-2j} = 1/(\rho^2 - 1)$ in both sides of (10), the optimization (10) is equivalent to

$$\min_{\{q_n\}_{n=0}^{\infty}} \sum_{j=1}^{\infty} \frac{\rho^{-2j}}{1 - \sum_{n=0}^{j-1} q_n} \quad \text{subject to} \quad \sum_{n=1}^{\infty} n q_n = N, \ \sum_{n=0}^{\infty} q_n = 1 \ \text{and} \ q_n \geq 0. \quad (16)$$

Note that the equality conditions can be written as

$$N = \sum_{n=1}^{\infty} n q_n = \sum_{n=1}^{\infty} \sum_{j=1}^{n} q_n = \sum_{j=1}^{\infty} \sum_{n=j}^{\infty} q_n = \sum_{j=1}^{\infty} \left( 1 - \sum_{n=0}^{j-1} q_n \right). \quad (17)$$

By Cauchy-Schwarz inequality for infinite series, we have

$$N \sum_{j=1}^{\infty} \frac{\rho^{-2j}}{1 - \sum_{n=0}^{j-1} q_n} = \left( \sum_{j=1}^{\infty} \left( 1 - \sum_{n=0}^{j-1} q_n \right) \right) \left( \sum_{j=1}^{\infty} \frac{\rho^{-2j}}{1 - \sum_{n=0}^{j-1} q_n} \right)$$

$$\geq \left( \sum_{j=1}^{\infty} \rho^{-j} \right)^2 = \frac{1}{(\rho - 1)^2}$$

and the equality holds when $q_0 = 1 - N(\rho-1)\rho^{-1}$ and $q_n = N(\rho-1)^2 \rho^{-(n+1)}$ for $n \geq 1$. However, this solution is not feasible when a given integer $N$ is greater than $\frac{\rho}{\rho-1}$ (due to $q_0 < 0$). The solution of (16) exists since the feasible region is closed and nonempty. For example,

$$q_n^* = \begin{cases} 0 & \text{for } 0 \leq n \leq k, \\ 1 - \dfrac{(N - k - 1)(\rho - 1)}{\rho} & \text{for } n = k+1, \\ \dfrac{(N - k - 1)(\rho - 1)^2}{\rho^{n-k}} & \text{for } k + 2 \leq n \end{cases} \quad (18)$$

with $k := N - 1 - \lfloor \frac{\rho}{\rho-1} \rfloor$ is feasible and achieves the objective function of (16)

$$\frac{1 - \rho^{-2(k+1)}}{\rho^2 - 1} + \frac{1}{(N-k-1)(\rho-1)^2 \rho^{2(k+1)}}.^2$$

To figure out that $q^*$ is the optimal solution, one can investigate KKT conditions of (16). However, in general, KKT theorem can not be applied to infinite dimensional problems. Instead, we consider the finite dimensional approximation of (16):

$$\min_{q_0, \ldots, q_T} \sum_{j=1}^{T} \frac{\rho^{-2j}}{1 - \sum_{n=0}^{j-1} q_n} \quad \text{subject to} \quad \sum_{n=0}^{T} n q_n = N, \sum_{n=0}^{T} q_n = 1 \quad \text{and} \quad q_n \geq 0. \qquad (19)$$

As we show in later, one can obtain the optimal solution of (19) for suffciently large $T$ using KKT conditions, which is

$$q_n = \begin{cases} 0 & \text{for } 0 \leq n \leq k, \\ 1 - \dfrac{(N-k-1)(\rho-1)}{1 - \rho^{-T+k+1}} \rho^{-1} & \text{for } n = k+1, \\ \dfrac{(N-k-1)(\rho-1)^2}{1 - \rho^{-T+k+1}} \rho^{-n+k} & \text{for } k+2 \leq n \leq T-1 \\ \dfrac{(N-k-1)(\rho-1)}{\rho^{T-k-1} - 1} & \text{for } n = T \end{cases} \qquad (20)$$

with $k := N - 1 - \lfloor \frac{\rho}{\rho-1} \rfloor$ and achieves the minimum

$$\frac{1 - \rho^{-2(k+1)}}{\rho^2 - 1} + \frac{\left(1 - \rho^{-T+k+1}\right)^2}{(N-k-1)(\rho-1)^2 \rho^{2(k+1)}}. \qquad (21)$$

We will show that the minimum of the infinite problem (16) is equivalent to the limit of (21) (a similar approach was introduced in [28]). We first extend $q_n$ to the point with infinite dimension.

Let $q_n^{(T)} = q_n$ for $n \leq T$ and $q_n^{(T)} = 0$ for $n > T$, then $q^{(T)} = (q_0^{(T)}, q_1^{(T)}, \ldots)$ is a feasible point of (16). Note that $\lim_{T \to \infty} q_n^{(T)} = q_n^*$ for all $n$. Define that

$$f(q, T) = \begin{cases} \displaystyle\sum_{j=1}^{T} \frac{\rho^{-2j}}{1 - \sum_{n=0}^{j-1} q_n} := C(q; T), & T = 1, 2, \ldots, \\ \displaystyle\sum_{j=1}^{\infty} \frac{\rho^{-2j}}{1 - \sum_{n=0}^{j-1} q_n} := C(q), & T = \infty \end{cases}$$

for $q = (q_0, q_1, \ldots)$. We claim that $f$ is continuous. Suppose $T_i \in \mathbb{N}$ is a nondecreasing infinite sequence such that $T_i > k, T_i \to \infty$ and $q^{(T_i)} \to q^*$ as $i \to \infty$. Consider that

$$\left| f(q^*, \infty) - f(q^{(T_i)}, T_i) \right| = \left| C(q^*) - C(q^{(T_i)}; T_i) \right|$$

$$= \left| \sum_{j=1}^{\infty} \frac{\rho^{-2j}}{1 - \sum_{n=0}^{j-1} q_n^*} - \sum_{j=1}^{T_i} \frac{\rho^{-2j}}{1 - \sum_{n=0}^{j-1} q_n^{(T_i)}} \right|$$

$$\leq \left| \sum_{j=T_i+1}^{\infty} \frac{\rho^{-2j}}{1 - \sum_{n=0}^{j-1} q_n^*} \right| + \left| \sum_{j=1}^{T_i} \rho^{-2j} \left( \frac{1}{1 - \sum_{n=0}^{j-1} q_n^*} - \frac{1}{1 - \sum_{n=0}^{j-1} q_n^{(T_i)}} \right) \right|$$

$$\leq \frac{\rho^{-T_i-k}}{(N-k-1)(\rho-1)} + \frac{\rho^{-T_i-k-1}}{(N-k-1)(\rho-1)^2} \qquad (22)$$

and (22) goes to zero as $i \to \infty$. In addition, the feasible set of (19) is nondecreasing, i.e., if we define the feasible regions as

$$X(T) := \left\{ q : \sum_{n=0}^{T} nq_n = N, \sum_{n=0}^{T} q_n = 1, q_n \geq 0, q_n = 0 \quad \text{for } n > T \right\},$$

$$X := \left\{ q : \sum_{n=0}^{\infty} nq_n = N, \sum_{n=0}^{\infty} q_n = 1, q_n \geq 0 \right\}$$

then $X(T) \subseteq X(T+1)$ for any $T$. This leads to $\lim_{T \to \infty} X(T) = \cup_{T \geq 1} X(T) = X$. Therefore, by the Berge's Maximum Theorem [3], the minimum of the finite dimensional problem (21) converges to that of infinite problem (16), i.e.,

$$\min \left\{ \sum_{j=1}^{\infty} \frac{\rho^{-2j}}{1 - \sum_{n=0}^{j-1} q_n} : q \in X \right\} = \lim_{T \to \infty} \min \left\{ \sum_{j=1}^{T} \frac{\rho^{-2j}}{1 - \sum_{n=0}^{j-1} q_n} : q \in X(T) \right\}$$

$$= \lim_{T \to \infty} \left( \frac{1 - \rho^{-2(k+1)}}{\rho^2 - 1} + \frac{\left(1 - \rho^{-T+k+1}\right)^2}{(N - k - 1)(\rho - 1)^2 \rho^{2(k+1)}} \right)$$

$$= \frac{1 - \rho^{-2(k+1)}}{\rho^2 - 1} + \frac{1}{(N - k - 1)(\rho - 1)^2 \rho^{2(k+1)}}.$$

Since $q^*$ in (18) achieves the above minimum, it follows that $q^*$ in (18) is the minimizer of (16).

The remaining part is to obtain the solution of the finite dimensional approximation (19) using KKT conditions. Since the objective and all inequality conditions are *convex* functions, any feasible solution that satisfies KKT conditions are optimal. Define the Lagrangian as

$$\mathcal{L}(q, \lambda, \nu) = \sum_{j=1}^{T} \frac{\rho^{-2j}}{1 - \sum_{n=0}^{j-1} q_n} + \lambda_1 \left( \sum_{n=0}^{T} nq_n - N \right) + \lambda_2 \left( \sum_{n=0}^{T} q_n - 1 \right) - \sum_{n=0}^{T} \nu_n q_n$$

where $\lambda_1, \lambda_2$ and $\nu_0, \ldots, \nu_T$ are the Lagrangian multipliers of equality and inequality condition, respectively. The corresponding KKT conditions are following:

- **Stationary:** For $0 \leq n \leq T$,

$$\frac{\partial \mathcal{L}}{\partial q_n} = \sum_{j=n+1}^{T} \frac{\rho^{-2j}}{(1 - \sum_{n'=0}^{j-1} q_{n'})^2} + \lambda_1 n + \lambda_2 - \nu_n = 0, \tag{C1}$$

- **Primal feasibility:**

$$\sum_{n=0}^{T} nq_n = N, \ \sum_{n=0}^{T} q_n = 1, \ q_n \geq 0, \tag{C2}$$

- **Dual feasibility:** For $0 \leq n \leq T$,

$$\nu_n \geq 0, \tag{C3}$$

- **Complementary slackness:** For $0 \leq n \leq T$,

$$\nu_n q_n = 0. \tag{C4}$$

Consider $(q, \lambda, \nu)$ that satisfies the KKT conditions holds that $\nu_n = 0$, $k + 1 \leq n \leq T$ and $\nu_n \neq 0$, $0 \leq n \leq k$ for some $k \in [0, T]$. By the complementary slackness (C4), $q_0 = q_1 = \cdots = q_k = 0$. Substracting two consecutive stationary conditions (C1), we obtain for $0 \leq n \leq T - 1$

$$\frac{\partial \mathcal{L}}{\partial q_n} - \frac{\partial \mathcal{L}}{\partial q_{n+1}} = \frac{\rho^{-2(n+1)}}{\left(1 - \sum_{n'=0}^{n} q_{n'}\right)^2} - \lambda_1 - \nu_n + \nu_{n+1} = 0, \tag{23}$$

which implies that

$$1 - \sum_{n'=0}^{n} q_{n'} = \frac{\rho^{-(n+1)}}{\sqrt{\lambda_1}} \quad \text{for } k+1 \leq n \leq T-1. \tag{24}$$

Putting them together into the equality condition (17) gives

$$N = \sum_{n=0}^{T-1} \left(1 - \sum_{n'=0}^{n} q_{n'}\right) = k+1 + \frac{1 - \rho^{-(T-k-1)}}{\sqrt{\lambda_1}\rho^{k+1}(\rho-1)},$$

equivalently, $\sqrt{\lambda_1} = \frac{1-\rho^{-(T-k-1)}}{(N-k-1)\rho^{k+1}(\rho-1)}$. Therefore, we obtain the solution from (24):

$$q_n = \begin{cases} 0 & \text{for } 0 \leq n \leq k, \\ 1 - \dfrac{(N-k-1)(\rho-1)}{1-\rho^{-T+k+1}}\rho^{-1} & \text{for } n = k+1, \\ \dfrac{(N-k-1)(\rho-1)^2}{1-\rho^{-T+k+1}}\rho^{-n+k} & \text{for } k+2 \leq n \leq T-1 \\ \dfrac{(N-k-1)(\rho-1)}{\rho^{T-k-1}-1} & \text{for } n = T \end{cases}$$

In order to satisfy the primal feasibility (C2), it should hold that

$$\frac{\rho(1-\rho^{-T+k+1})}{\rho-1} \geq N-k-1 \quad \text{and} \quad k \leq N-1. \tag{25}$$

From (23), the dual variables $\nu$ can be written as for $n \leq k$

$$\nu_n - \nu_{n+1} = \rho^{-2(n+1)} - \lambda_1$$

and in order to satisfy the dual feasibility (C3), i.e., $\nu_n > 0$ for $n \leq k$, the sufficient condition is

$$N-k-1 > \frac{(1-\rho^{-T+k+1})}{\rho-1}. \tag{26}$$

To satisfy both (25) and (26), there exists an integer in the interval $\left[\frac{(1-\rho^{-T+k+1})}{\rho-1}, \frac{\rho(1-\rho^{-T+k+1})}{\rho-1}\right]$. We now choose $T$ large enough such that

$$\left\lfloor \frac{\rho}{\rho-1} \right\rfloor \leq \frac{\rho(1-\rho^{-T+N})}{\rho-1},$$

and it holds that $\left\lfloor \frac{\rho}{\rho-1} \right\rfloor \in \left[\frac{(1-\rho^{-T+k+1})}{\rho-1}, \frac{\rho(1-\rho^{-T+k+1})}{\rho-1}\right]$ for some $0 \leq k \leq N-1$. By choosing $k := N-1-\left\lfloor \frac{\rho}{\rho-1} \right\rfloor$, $\{q_n\}_{n=0}^{T}$ in (20) satisfies the KKT conditions and acheives the minimum

$$\frac{1-\rho^{-2(k+1)}}{\rho^2-1} + \frac{\left(1-\rho^{-T+k+1}\right)^2}{(N-k-1)(\rho-1)^2\rho^{2(k+1)}}.$$

### B.3 Proof of Theorem 5 : convergence analysis of SGD

We recall that $\theta^{(t)} \in \mathcal{C} \subseteq \mathbb{R}^{d'}$ by the parameter in the $t$-th iteration and $\theta_i^{(t)}$ by its element $i$-th position for $i = 1, \ldots, d'$. For simplicity, we denote that

$$h(\theta) := \Sigma_f(A(\theta)) + g(\theta)$$

and $\theta^* \in \mathcal{C}$ be the optimal of $h$. Let $\psi^{(t)}$ be our unbiased gradient estimator for $\Sigma_f(A(\theta))$ using $\{\mathbf{v}^{(k)}\}_{k=1}^{M}$ and $n$, that is,

$$\mathbf{E}_{n,\mathbf{v}}[\psi^{(t)}] = \frac{\partial}{\partial \theta}\Sigma_f(A(\theta))$$

and $\nabla g^{(t)}$ be the derivative of $g(\theta)$ at $\theta^{(t)}$. Unless stated otherwise, we use $\|\cdot\|$ as the entry-wise $L_2$-norm, i.e., $L_2$-norm for vectors and Frobenius norm for matrices. Now we are ready to show the convergence guarantee for SGD. The iteration of SGD can be written as

$$\theta^{(t+1)} = \Pi_{\mathcal{C}}\left(\theta^{(t)} - \eta(\psi^{(t)} + \nabla g^{(t)})\right)$$

where $\Pi_{\mathcal{C}}(\cdot)$ is the projection mapping in $\mathcal{C}$. The remaining part is similar with standard proof of the projected stochastic gradient descent. First, we write the error between $\theta^{(t)}$ and $\theta^*$ as

$$
\begin{aligned}
\|\theta^{(t+1)} - \theta^*\|^2 &= \|\Pi_{\mathcal{C}}(\theta^{(t)} - \eta(\psi^{(t)} + \nabla g^{(t)})) - \theta^*\|^2 \\
&\leq \|\theta^{(t)} - \eta(\psi^{(t)} + \nabla g^{(t)}) - \theta^*\|^2 \\
&= \|\theta^{(t)} - \theta^*\|^2 - 2\eta\left\langle \psi^{(t)} + \nabla g^{(t)}, \theta^{(t)} - \theta^* \right\rangle + \eta^2 \|\psi^{(t)} + \nabla g^{(t)}\|^2 \\
&\leq \|\theta^{(t)} - \theta^*\|^2 - 2\eta\left\langle \psi^{(t)} + \nabla g^{(t)}, \theta^{(t)} - \theta^* \right\rangle + 2\eta^2\|\psi^{(t)}\|^2 + 2\eta^2\|\nabla g^{(t)}\|^2 \\
&\leq \|\theta^{(t)} - \theta^*\|^2 - 2\eta\left\langle \psi^{(t)} + \nabla g^{(t)}, \theta^{(t)} - \theta^* \right\rangle + 2\eta^2\|\psi^{(t)}\|^2 + 2\eta^2 L_g^2
\end{aligned}
$$

where the inequality in the second line holds from the convexity of $\mathcal{C}$, the inequality in the fourth line follows from that $\|a + b\|^2 \leq 2\|a\|^2 + 2\|b\|^2$ and the last inequality follows from Lipschitz continuity of $g$. Taking the expectation with respect to random samples (i.e., random degree and vectors) in $t$-th iteration, which denoted as $\mathbf{E}_t[\cdot]$, we have

$$\mathbf{E}_t[\|\theta^{(t+1)} - \theta^*\|^2] \leq \|\theta^{(t)} - \theta^*\|^2 - 2\eta\left\langle \nabla h(\theta^{(t)}), \theta^{(t)} - \theta^* \right\rangle + 4\eta^2 B^2 \qquad (27)$$

where $B^2 := \max\left(\mathbf{E}_t[\|\psi^{(t)}\|^2], L_g^2\right)$. In addition, by $\alpha$-strong convexity of $h$, it holds that

$$\alpha\|\theta^{(t)} - \theta^*\|^2 \leq \left\langle \nabla h(\theta^{(t)}), \theta^{(t)} - \theta^* \right\rangle. \qquad (28)$$

Combining (27) with (28) and taking the expectation on both sides with respect to all random samples from $1, ..., t$ iteration, we obtain that

$$\mathbf{E}[\|\theta^{(t+1)} - \theta^*\|^2] \leq (1 - 2\eta\alpha)\mathbf{E}[\|\theta^{(t)} - \theta^*\|^2] + 4\eta^2 B^2$$

Applying $\eta = \frac{1}{\alpha t}$, we have

$$\mathbf{E}[\|\theta^{(t+1)} - \theta^*\|^2] \leq \left(1 - \frac{2}{t}\right)\mathbf{E}[\|\theta^{(t)} - \theta^*\|^2] + \frac{4B^2}{\alpha^2 t^2}.$$

Therefore, if $\mathbf{E}[\|\theta^{(1)} - \theta^*\|^2] \leq 4B^2/\alpha^2$ holds, then the result follows by induction on $t \geq 1$. Under assumption that $\mathbf{E}[\|\theta^{(t)} - \theta^*\|^2] \leq 4B^2/(\alpha^2 t)$, it is straightforward that

$$\mathbf{E}[\|\theta^{(t+1)} - \theta^*\|^2] \leq \left(1 - \frac{2}{t}\right)\frac{4B^2}{\alpha^2 t} + \frac{4B^2}{\alpha^2 t^2} \leq \frac{4B^2}{\alpha^2}\left(\frac{1}{t+1}\right).$$

To show the case of $t = 1$, we recall the strong convexity of $h$ and use Cauchy-Schwartz inequality:

$$\alpha\|\theta^{(1)} - \theta^*\|^2 \leq \left\langle \psi^{(1)} + \nabla g^{(1)}, \theta^{(1)} - \theta^* \right\rangle \leq \|\psi^{(1)} + \nabla g^{(1)}\|\|\theta^{(1)} - \theta^*\|,$$

which leads to that

$$\alpha^2 \mathbf{E}[\|\theta^{(1)} - \theta^*\|^2] \leq \mathbf{E}[\|\psi^{(1)} + \nabla g^{(1)}\|^2] \leq 4B^2.$$

Recall that Lemma 4 implies that for all $t$

$$\mathbf{E}_t[\|\psi^{(t)}\|^2] \leq \left(2L_A^2/M + d'L_{\text{nuc}}^2\right)\left(C_1 + C_2 N^4 \rho^{-2N}\right).$$

for some constants $C_1, C_2 > 0$. This completes the proof of Theorem 5.

## B.4    Proof of Theorem 6 : convergence analysis of SVRG

Denote the objective as $h(\theta) := \Sigma_f(A(\theta)) + g(\theta)$. Let $\psi^{(t)}, \widetilde{\psi}$ be our unbiased gradient estimator for $\Sigma_f(A(\theta))$ at $\theta^{(t)}$ and $\widetilde{\theta}^{(s)}$, respectively, and $\widetilde{\mu} = \nabla\Sigma_f(A(\widetilde{\theta}^{(s)}))$. We use $\nabla g^{(t)}$ by the exact gradient of $g(\theta)$ at $\theta^{(t)}$, which is easy to compute. The iteration of SVRG can be written as

$$\theta^{(t+1)} = \Pi_{\mathcal{C}}(\theta^{(t)} - \eta\xi^{(t)}), \quad \text{where} \quad \xi^{(t)} := \psi^{(t)} - \widetilde{\psi} + \widetilde{\mu} + \nabla g^{(t)}$$

where $\Pi_{\mathcal{C}}(\cdot)$ is the projection mapping in $\mathcal{C}$. We first introduce the lemma that implies our unbiased estimator is $\beta$-smooth for some $\beta > 0$.

**Lemma 8** *Suppose that assumptions $(\mathcal{A}0)$-$(\mathcal{A}2)$ hold and assume that $A : \mathcal{C} \to \mathcal{S}^{d \times d}$ is $\beta_A$-smooth function with respect to $\|\cdot\|_F$. Let $\psi, \psi'$ be our unbiased gradient estimator (12) at $\theta, \theta' \in \mathcal{C} \subseteq \mathbb{R}$ using the same $\{\mathbf{v}^{(k)}\}_{k=1}^M$ and $n$ (drawn from (11) with mean $N$). Then, it holds that*

$$\mathbf{E}_{n,\mathbf{v}}\left[\|\psi + \nabla g(\theta) - \psi' - \nabla g(\theta')\|_2^2\right] \leq \left(2\beta_g^2 + \left(\frac{L_A^4 + \beta_A^2}{M} + L_A^4\right)\left(D_1 + \frac{D_2 N^8}{\rho^{2N}}\right)\right)\|\theta - \theta'\|_2^2.$$

*where $D_1, D_2 > 0$ are some constants independent of $M, N$.*

The proof of the above lemma is given in Section B.5. For notational simplicity, we denote

$$\beta^2 := 2\beta_g^2 + \left(\frac{L_A^4 + \beta_A^2}{M} + L_A^4\right)\left(D_1 + \frac{D_2 N^8}{\rho^{2N}}\right).$$

The remaining part mimics the analysis of [10]. Using the above lemma, the moment of the gradient estimator is bounded as

$$
\begin{aligned}
\mathbf{E}_t[\|\psi^{(t)} - \widetilde{\psi} + \widetilde{\mu} + \nabla g^{(t)}\|^2] &\leq 2\mathbf{E}_t[\|\psi^{(t)} + \nabla g^{(t)} - \psi^* - \nabla g^*\|^2] + 2\mathbf{E}_t[\|\widetilde{\psi} - \psi^* - \nabla g^* - \widetilde{\mu}\|^2] \\
&\leq 2\mathbf{E}_t[\|\psi^{(t)} + \nabla g^{(t)} - \psi^* - \nabla g^*\|^2] + 2\mathbf{E}_t[\|\widetilde{\psi} + \nabla\widetilde{g} - \psi^* - \nabla g^*\|^2] \\
&\leq 2\beta^2\left(\|\theta^{(t)} - \theta^*\|^2 + \|\widetilde{\theta} - \theta^*\|^2\right) \qquad (29)
\end{aligned}
$$

where the inequality in the first line holds from $\|a + b\|^2 \leq 2(\|a\|^2 + \|b\|^2)$, the inequality in the second line holds that $\mathbf{E}[\|X - \mathbf{E}[X]\|^2] \leq \mathbf{E}[\|X\|^2]$ for any random variable $X$ and the last inequality holds from Lemma 8.

Now, we use similar procedures of Theorem 5 to obtain

$$
\begin{aligned}
\|\theta^{(t+1)} - \theta^*\|^2 &= \|\Pi_{\mathcal{C}}\left(\theta^{(t)} - \eta\xi^{(t)}\right) - \theta^*\|^2 \\
&\leq \|\theta^{(t)} - \eta\xi^{(t)} - \theta^*\|^2 \\
&= \|\theta^{(t)} - \theta^*\|^2 - 2\eta\left\langle\theta^{(t)} - \theta^*, \xi_t\right\rangle + \|\xi_t\|^2.
\end{aligned}
$$

where the inequality holds from the convexity of $\mathcal{C}$. Taking the expectation with respect to random samples of $t$-th iteration, which denoted as $\mathbf{E}_t[\cdot]$, we obtain that

$$
\begin{aligned}
\mathbf{E}_t[\|\theta^{(t+1)} - \theta^*\|^2] &= \|\theta^{(t)} - \theta^*\|^2 - 2\eta\left\langle\theta^{(t)} - \theta^*, \nabla h(\theta^{(t)})\right\rangle + \eta^2\mathbf{E}_t[\|\xi_t\|^2] \\
&\leq \|\theta^{(t)} - \theta^*\|^2 - 2\eta\alpha\|\theta^{(t)} - \theta^*\|^2 + \eta^2\mathbf{E}_t[\|\xi_t\|^2] \\
&\leq \|\theta^{(t)} - \theta^*\|^2 - 2\eta\alpha\|\theta^{(t)} - \theta^*\|^2 + 2\eta^2\beta^2\left(\|\theta^{(t)} - \theta^*\|^2 + \|\widetilde{\theta} - \theta^*\|^2\right)
\end{aligned}
$$

where the inequality in the second line holds from the $\alpha$-strong convexity of the objective and the last inequality holds from (29). Taking the expectation over the randomness of all iterations, we have

$$\mathbf{E}[\|\theta^{(t+1)} - \theta^*\|^2] - \mathbf{E}[\|\theta^{(t)} - \theta^*\|^2] \leq 2\eta\left(\eta\beta^2 - \alpha\right)\mathbf{E}[\|\theta^{(t)} - \theta^*\|^2] + 2\eta^2\beta^2\mathbf{E}[\|\widetilde{\theta} - \theta^*\|^2]$$

Summing both sides over $t = 1, 2, \ldots, T$, it yields that

$$\mathbf{E}[\|\theta^{(T)} - \theta^*\|^2] - \mathbf{E}[\|\theta^{(0)} - \theta^*\|^2] \leq 2\eta\left(\eta\beta^2 - \alpha\right)\sum_{t=0}^{T-1}\mathbf{E}[\|\theta^{(t)} - \theta^*\|^2] + 2T\eta^2\beta^2\mathbf{E}[\|\widetilde{\theta} - \theta^*\|^2]$$

Rearranging and using the facts that $\mathbf{E}[\|\theta^{(T)} - \theta^*\|^2] \geq 0$ and $\widetilde{\theta} = \widetilde{\theta}^{(s)}$, we get

$$2\eta\left(\alpha - \eta\beta^2\right)\sum_{t=0}^{T-1}\mathbf{E}[\|\theta^{(t)} - \theta^*\|^2] \leq \left(1 + 2T\eta^2\beta^2\right)\mathbf{E}[\|\theta^{(0)} - \theta^*\|^2].$$

From $\widetilde{\theta}^{(s+1)} = \frac{1}{T}\sum_{t=1}^{T}\theta^{(t)}$ and Jensen's inequality, we have

$$\mathbf{E}[\|\widetilde{\theta}^{(s+1)} - \theta^*\|^2] \leq \frac{1}{T}\sum_{t=1}^{T}\mathbf{E}[\|\theta^{(t)} - \theta^*\|^2] \leq \frac{1 + 2T\eta^2\beta^2}{2\eta T\left(\alpha - \eta\beta^2\right)}\mathbf{E}[\|\widetilde{\theta}^{(s)} - \theta^*\|^2]$$

Substituting $\eta = \frac{\alpha}{7\beta^2}$ and $T \geq \frac{49\beta^2}{2\alpha^2}$, we have that

$$\mathbf{E}[\|\widetilde{\theta}^{(S)} - \theta^*\|^2] \leq r^S\mathbf{E}[\|\widetilde{\theta}^{(0)} - \theta^*\|^2]$$

for some $0 < r < 1$.

### B.5    Proof of lemmas

#### B.5.1    Proof of Lemma 1

Without loss of generality, we choose $a = -1, b = 1$. An analytic function $f$ has an (unique) infinite Chebyshev series expansion: $f(x) = \sum_{j=0}^{\infty} b_j T_j(x)$. and recall that our proposed estimator as

$$\widehat{p}_n(x) = \sum_{j=0}^{n}\frac{b_j}{1 - \sum_{i=0}^{j-1} q_i}T_j(x).$$

To prove that $\mathbf{E}_n[\widehat{p}_n(x)] = f(x)$, we define two sequences:

$$A_M := \sum_{j=0}^{M}\sum_{n=j}^{M} q_n\frac{b_j T_j(x)}{1 - \sum_{i=0}^{j-1} q_i}, \quad B_{M,K} := \sum_{j=0}^{M}\sum_{n=j}^{K} q_n\frac{b_j T_j(x)}{1 - \sum_{i=0}^{j-1} q_i}.$$

Then, it is easy to show that

$$\lim_{M\to\infty} A_M = \sum_{j=0}^{\infty}\sum_{n=j}^{\infty} q_n\frac{b_j T_j(x)}{1 - \sum_{i=0}^{j-1} q_i} = \sum_{n=0}^{\infty} q_n\left(\sum_{j=0}^{n}\frac{b_j T_j(x)}{1 - \sum_{i=0}^{j-1} q_i}\right) = \sum_{n=0}^{\infty} q_n\widehat{p}_n(x) = \mathbf{E}_n[\widehat{p}_n(x)],$$

and

$$\lim_{M\to\infty}\lim_{K\to\infty} B_{M,K} = \lim_{M\to\infty}\sum_{j=0}^{M}\left(\sum_{n=j}^{\infty} q_n\right)\frac{b_j T_j(x)}{1 - \sum_{i=0}^{j-1} q_i} = \lim_{M\to\infty}\sum_{j=0}^{M} b_j T_j(x) = f(x).$$

In general, $A_M$ and $B_{M,K}$ might not converge to the same values. Now, consider sufficiently large $K \geq M$. From the condition that $\lim_{n\to\infty}\sum_{i=n+1}^{\infty} q_i\widehat{p}_n(x)$, we have

$$\mathbf{E}_n[\widehat{p}_n(x)] - f(x) = \lim_{M\to\infty}\lim_{K\to\infty}(A_M - B_{M,K}) = \lim_{M\to\infty}\lim_{K\to\infty}\left(\sum_{j=0}^{M}\sum_{n=M+1}^{K} q_n\frac{b_j T_j(x)}{1 - \sum_{i=0}^{j-1} q_i}\right)$$

$$= \lim_{M\to\infty}\lim_{K\to\infty}\left(\sum_{n=M+1}^{K} q_n\right)\left(\sum_{j=0}^{M}\frac{b_j T_j(x)}{1 - \sum_{i=0}^{j-1} q_i}\right)$$

$$= \lim_{M\to\infty}\left(\sum_{n=M+1}^{\infty} q_n\right)\left(\sum_{j=0}^{M}\frac{b_j T_j(x)}{1 - \sum_{i=0}^{j-1} q_i}\right)$$

$$= \lim_{M\to\infty}\left(\sum_{n=M+1}^{\infty} q_n\right)\widehat{p}_M(x) = 0.$$

Therefore, we can conclude that $\widehat{p}_n(x)$ is an unbiased estimator of $f(x)$. In addition, this also holds for the trace of matrices due to its linearity: $\mathbf{E}_n\left[\texttt{tr}\left(\widehat{p}_n(A)\right)\right] = \texttt{tr}\left(f(A)\right)$. By taking expectation over Rademacher random vectors $\mathbf{v}$ and degree $n$, we establish the unbiased estimator of spectral-sums:

$$\mathbf{E}_{n,\mathbf{v}}\left[\mathbf{v}^\top \widehat{p}_n(A)\,\mathbf{v}\right] = \mathbf{E}_n\left[\mathbf{E}_\mathbf{v}\left[\mathbf{v}^\top \widehat{p}_n(A)\,\mathbf{v}|n\right]\right] = \mathbf{E}_n\left[\texttt{tr}\left(\widehat{p}_n(A)\right)\right] = \texttt{tr}\left(f(A)\right),$$

For fixed $\mathbf{v}$ and $n$, the function $h(\theta) := \mathbf{v}^\top \hat{p}_n(A(\theta))\mathbf{v}$ is a linear combination of all entries of $A$, so the fact that all partial derivatives $\partial A_{j,k}/\partial \theta_i$ exist and are continuous implies that the partial derivatives of $h$ with respect to $\theta_1, \ldots, \theta_{d'}$ exist and are continuous. In particular, since expectation over $\mathbf{v} \in [-1, +1]^d$ is a finite sum, it is straightforward that the gradient operator and expectation operator can be interchanged:

$$\nabla_\theta \texttt{tr}\left(f(A)\right) = \nabla_\theta \mathbf{E}\left[\mathbf{v}^\top \widehat{p}_n(A)\,\mathbf{v}\right] = \mathbf{E}\left[\nabla_\theta \mathbf{v}^\top \widehat{p}_n(A)\,\mathbf{v}\right].$$

In the case of trace probing vector $\mathbf{v}$ is a continuous random vector, i.e., Gaussian, we turn to use the Leibniz rule which allows to interchange the gradient operator and expectation operator. Hence, we conclude the same result. This completes the proof of Lemma 1.

### B.5.2  Proof of Lemma 2

Without loss of generality, we choose $a = -1, b = 1$. We first introduce the orthogonality of Chebyshev polynomials of the first kind, that is,

$$\int_{-1}^{1} \frac{T_i(x)T_j(x)}{\sqrt{1-x^2}}\,dx = \begin{cases} 0 & i \neq j, \\ \pi & i = j = 0, \\ \frac{\pi}{2} & i = j \neq 0. \end{cases}$$

Given functions $f, g$ defined on $[-1, 1]$, Chebyshev induced inner-product and weighted norm are defined as

$$\langle f, g\rangle_C = \int_{-1}^{1} \frac{f(x)g(x)}{\sqrt{1-x^2}}\,dx, \qquad \|f\|_C^2 = \langle f, f\rangle_C.$$

For a fixed $n$, the square of Chebyshev weighted error can be written as

$$
\begin{aligned}
\|\widehat{p}_n - f\|_C^2 &= \|\widehat{p}_n - p_n + p_n - f\|_C^2 = \|p_n - f\|_C^2 + 2\langle p_n - f, \widehat{p}_n - p_n\rangle_C + \|\widehat{p}_n - p_n\|_C^2 \\
&\overset{(\dagger)}{=} \|p_n - f\|_C^2 + \|\widehat{p}_n - p_n\|_C^2 \\
&= \left\|\sum_{j=n+1}^{\infty} b_j T_j\right\|_C^2 + \left\|\sum_{j=1}^{n} \frac{\sum_{k=0}^{j-1} q_n}{1 - \sum_{k=0}^{j-1} q_n} b_j T_j\right\|_C^2 \\
&\overset{(\ddagger)}{=} \frac{\pi}{2} \sum_{j=n+1}^{\infty} b_j^2 + \frac{\pi}{2} \sum_{j=1}^{n} \left(\frac{\sum_{i=0}^{j-1} q_i}{1 - \sum_{i=0}^{j-1} q_i} b_j\right)^2.
\end{aligned}
$$

Both the second equality ($\dagger$) and the last equality ($\ddagger$) come from the orthogonality of Chebyshev polynomials and the following facts:

$$
\begin{aligned}
p_n - f &: \text{linear combination of } T_{n+1}(x), T_{n+2}(x), \cdots, \\
\widehat{p}_n - p_n &: \text{linear combination of } T_0(x), \cdots, T_n(x).
\end{aligned}
$$

The Chebyshev weighted variance can be computed by taking expectation with respect to $n$:

$$\frac{2}{\pi}\mathbf{E}_n[\|\widehat{p}_n - f\|_C^2] = \frac{2}{\pi}\sum_{n=0}^{\infty} q_n\|\widehat{p}_n - f\|_C^2 = q_0\sum_{j=1}^{\infty} b_j^2 + \sum_{n=1}^{\infty} q_n\left(\sum_{j=1}^{n}\left(\frac{b_j\sum_{i=0}^{j-1} q_i}{1 - \sum_{i=0}^{j-1} q_i}\right)^2 + \sum_{j=n+1}^{\infty} b_j^2\right)$$

$$= \sum_{j=1}^{\infty} b_j^2\left(q_0 + \sum_{i=1}^{j-1} q_i + \left(\frac{\sum_{i=0}^{j-1} q_i}{1 - \sum_{i=0}^{j-1} q_i}\right)^2\sum_{i=j}^{\infty} q_i\right)$$

$$= \sum_{j=1}^{\infty} b_j^2\left(\sum_{i=0}^{j-1} q_i + \left(\frac{\sum_{i=0}^{j-1} q_i}{1 - \sum_{i=0}^{j-1} q_i}\right)^2\left(1 - \sum_{i=0}^{j-1} q_i\right)\right)$$

$$= \sum_{j=1}^{\infty} b_j^2\left(\sum_{i=0}^{j-1} q_i + \frac{\left(\sum_{i=0}^{j-1} q_i\right)^2}{1 - \sum_{i=0}^{j-1} q_i}\right) = \sum_{j=1}^{\infty} b_j^2\left(\frac{\sum_{i=0}^{j-1} q_i}{1 - \sum_{i=0}^{j-1} q_i}\right).$$

This completes the proof of Lemma 2.

### B.5.3   Proof of Lemma 4

First, we define the $n$ degree Chebyshev polynomials of the first kind by $T_n(\cdot)$ and the second kind by $U_n(\cdot)$. One important property is that $T_n'(x) := \frac{d}{dx}T_n(x) = nU_{n-1}(x)$ for $n \geq 1$ (see [20]). Consider our unbiased estimator with a single random sample, i.e., a Rademacher vector $\mathbf{v}$ and a degree $n$ drawn from the optimal distribution (11).

From the intermediate result (34) in the proof of Lemma 7, the gradient estimator can be written as following:

$$\psi_i := \frac{\partial}{\partial\theta_i}\mathbf{v}^\top\widehat{p}_n\left(A(\theta)\right)\mathbf{v} = \frac{2}{b-a}\mathbf{v}^\top G\mathbf{v} \tag{30}$$

where

$$G = \sum_{j=0}^{n-1}\widehat{b}_{j+1}\left(2\sum_{r=0}^{j}{}'T_r(\widetilde{A})\frac{\partial A}{\partial\theta_i}U_{j-r}(\widetilde{A})\right)$$

and $\widetilde{A} = \frac{2}{b-a}A(\theta) - \frac{b+a}{b-a}I$ and $\sum'$ implies the summation where the first term is halved. We also note that $\mathtt{tr}\,(G) = \mathtt{tr}\left(\frac{\partial A}{\partial\theta_i}\widehat{p}_n'(A)\right)$. Here, our goal is to find the upper bound of $\mathbf{E}_{n,\mathbf{v}}[\psi_i^2]$, that is,

$$\frac{(b-a)^2}{4}\mathbf{E}_{n,\mathbf{v}}[\psi_i^2] = \mathbf{E}_{n,\mathbf{v}}\left[\left(\mathbf{v}^\top G\mathbf{v}\right)^2\right] = \mathbf{E}_n\left[\mathbf{E}_\mathbf{v}\left[\left(\mathbf{v}^\top G\mathbf{v}\right)^2\big|n\right]\right].$$

From [14, 2], we have that $\mathrm{Var}_\mathbf{v}[\mathbf{v}^\top A\mathbf{v}] = 2(\|A\|_F^2 - \sum_{i=1}^{d} A_{ii}^2) \leq 2\|A\|_F^2$ and $\mathbf{E}_\mathbf{v}[\mathbf{v}^\top G\mathbf{v}] = \mathtt{tr}\,(G)$ for Rademacher random vector $\mathbf{v} \in [-1, 1]^d$ and $A \in \mathcal{S}^{d\times d}$. Therefore, we have

$$\mathbf{E}_\mathbf{v}\left[\left(\mathbf{v}^\top G\mathbf{v}\right)^2\big|n\right] = \mathrm{Var}_\mathbf{v}[\mathbf{v}^\top G\mathbf{v}|n] + \mathbf{E}_\mathbf{v}\left[\mathbf{v}^\top G\mathbf{v}|n\right]^2 \leq 2\|G\|_F^2 + \left(\mathtt{tr}\,(G)\right)^2. \tag{31}$$

The first term in (31) is bounded as

$$2\|G\|_F^2 \leq 2\left\|\frac{\partial A}{\partial\theta_i}\right\|_F^2\left(\sum_{j=1}^{n}\left|\widehat{b}_j\right|\left(2\sum_{r=0}^{j}{}'\|T_r(\widetilde{A})\|_2\|U_{j-r}(\widetilde{A})\|_2\right)\right)^2$$

$$\leq 2\left\|\frac{\partial A}{\partial\theta_i}\right\|_F^2\left(\sum_{j=1}^{n}\left|\widehat{b}_j\right|\left(2\sum_{r=0}^{j}{}'(j-r+1)\right)\right)^2$$

$$= 2\left\|\frac{\partial A}{\partial\theta_i}\right\|_F^2\left(\sum_{j=1}^{n}\left|\widehat{b}_j\right|j^2\right)^2$$

which the first inequality comes from the triangle inequality of $\|\cdot\|_F$ and the fact that $\|XY\|_F \leq \|X\|_F \|Y\|_2$ for mutliplicable matrices $X$ and $Y$. The inequality in the second line holds from $\|T_i(\widetilde{A})\|_2 \leq 1$ and $\|U_i(\widetilde{A})\|_2 \leq i+1$ for $i \geq 0$.

For second term in (31), we use the inequality that $\mathrm{tr}\,(XY) \leq \|X\|_{\mathrm{nuc}} \|Y\|_2$ for real symmetric matrices $X, Y$ (see Section B.5.8) to obtain

$$
\left(\mathrm{tr}\,(G)\right)^2 = \left(\mathrm{tr}\left(\frac{\partial A}{\partial \theta_i} \widehat{p}'_n(A)\right)\right)^2 \leq \left\|\frac{\partial A}{\partial \theta_i}\right\|^2_{\mathrm{nuc}} \|\widehat{p}'_n(A)\|^2_2
$$

$$
= \left\|\frac{\partial A}{\partial \theta_i}\right\|^2_{\mathrm{nuc}} \left\|\sum_{j=1}^n \widehat{b}_j j U_{j-1}(\widetilde{A})\right\|^2_2
$$

$$
\leq \left\|\frac{\partial A}{\partial \theta_i}\right\|^2_{\mathrm{nuc}} \left(\sum_{j=1}^n \left|\widehat{b}_j\right| j^2\right)^2
$$

where the equality in the second line uses that $\left(\sum_{j=0}^n \widehat{b}_j T_j(x)\right)' = \sum_{j=1}^n \widehat{b}_j j U_{j-1}(x)$ and the last inequality holds from $\|U_i(\widetilde{A})\|_2 \leq i+1$. Putting all together into (31) and summing for all $i = 1, \ldots, d'$, we obtain that

$$
\mathbf{E}_{n,\mathbf{v}}[\psi^2] = \sum_{i=1}^{d'} \mathbf{E}_{n,\mathbf{v}}[\psi_i^2] \leq \frac{4}{(b-a)^2} \sum_{i=1}^{d'} \mathbf{E}_n \left[2\|G\|_F^2 + (\mathrm{tr}\,(G))^2\right]
$$

$$
\leq \frac{4}{(b-a)^2} \sum_{i=1}^{d'} \left(2\left\|\frac{\partial A}{\partial \theta_i}\right\|^2_F + \left\|\frac{\partial A}{\partial \theta_i}\right\|^2_{\mathrm{nuc}}\right) \mathbf{E}_n \left[\left(\sum_{j=1}^n \left|\widehat{b}_j\right| j^2\right)^2\right]
$$

$$
\leq \frac{4}{(b-a)^2} \left(2\left\|\frac{\partial A}{\partial \theta}\right\|^2_F + \sum_{k=1}^{d'} \left\|\frac{\partial A}{\partial \theta_i}\right\|^2_{\mathrm{nuc}}\right) \mathbf{E}_n \left[\left(\sum_{j=1}^n \left|\widehat{b}_j\right| j^2\right)^2\right].
$$

When we estimate $\psi$ using $M$ Rademacher random vectors $\{\mathbf{v}^{(k)}\}_{k=1}^M$, the variance in (31) is reduced by $1/M$. Hence, we have

$$
\mathbf{E}_{n,\mathbf{v}}[\psi^2] \leq \frac{4}{(b-a)^2} \left(\frac{2}{M}\left\|\frac{\partial A}{\partial \theta}\right\|^2_F + \sum_{k=1}^{d'} \left\|\frac{\partial A}{\partial \theta_i}\right\|^2_{\mathrm{nuc}}\right) \mathbf{E}_n \left[\left(\sum_{j=1}^n \left|\widehat{b}_j\right| j^2\right)^2\right]
$$

$$
\leq \frac{4}{(b-a)^2} \left(\frac{2L_A^2}{M} + d' L_{\mathrm{nuc}}^2\right) \mathbf{E}_n \left[\left(\sum_{j=1}^n \left|\widehat{b}_j\right| j^2\right)^2\right].
$$

Finally, we introduce the following lemma to bound the right-hand side, where its proof is given in Section B.5.6.

**Lemma 9** *Suppose that $q_n^*$ is the optimal degree distribution as defined in (11) and $b_j$ is the Chebyshev coefficients of analytic function $f$. Define the weighted coefficient $\widehat{b}_j$ as $\widehat{b}_j = b_j/(1 - \sum_{i=0}^{j-1} q_i^*)$ for $j \geq 0$ and conventionally $q_{-1}^* = 0$. Then, there exists constants $C_1, C_2 > 0$ independent of $M, N$ such that*

$$
\sum_{n=1}^\infty q_n^* \left(\sum_{j=1}^n |\widehat{b}_j| j^2\right)^2 \leq C_1 + \frac{C_2 N^4}{\rho^{2N}}.
$$

To sum up, we conclude that

$$
\mathbf{E}_{n,\mathbf{v}}[\psi^2] \leq \left(\frac{2L_A^2}{M} + d' L_{\mathrm{nuc}}^2\right) \left(C_1 + \frac{C_2 N^4}{\rho^{2N}}\right)
$$

for some constant $C_1, C_2 > 0$. This completes the proof of Lemma 4.

### B.5.4   Proof of Lemma 7

We consider more general case in which $A \in \mathcal{S}^{d \times d}$ is a function of parameter $\theta = [\theta_1, \ldots, \theta_{d'}]$, and our goal is to derive a closed form of $\frac{\partial}{\partial \theta_i} \mathbf{v}^\top p_n(A) \mathbf{v}$ with allowing only vector operations. We begin by observing that for any polynomial $p_n$ and symmetric matrix $A \in \mathcal{S}^{d \times d}$, the derivative of $\mathbb{E}_{\mathbf{v}}[\mathbf{v}^\top p_n(A) \mathbf{v}]$ can be expressed by a simple formulation, that is,

$$\frac{\partial}{\partial \theta_i} \mathbb{E}_{\mathbf{v}}[\mathbf{v}^\top p_n(A) \mathbf{v}] = \frac{\partial}{\partial \theta_i} \mathrm{tr}(p_n(A)) = p_n'(A) \frac{\partial A}{\partial \theta_i}.$$

However, it does not holds that

$$\frac{\partial}{\partial \theta_i} \mathbf{v}^\top p_n(A) \mathbf{v} = \frac{\partial}{\partial \theta_i} \mathrm{tr}\left(p_n(A) \mathbf{v} \mathbf{v}^\top\right) \neq p_n'(A) \mathbf{v} \mathbf{v}^\top \frac{\partial A}{\partial \theta_i}.$$

for some vector $\mathbf{v} \in \mathbb{R}^d$. This is because of $\frac{\partial}{\partial \theta_i} \mathrm{tr}\left(A^j \mathbf{v} \mathbf{v}^\top\right) \neq j A^{j-1} \mathbf{v} \mathbf{v}^\top \frac{\partial A}{\partial \theta_i}$ in general.

If $p_n(x)$ is the truncated Chebyshev series, i.e., $p_n(x) = \sum_{j=0}^n b_j T_j(x)$, we can compute $\frac{\partial}{\partial \theta_i} \mathbf{v}^\top p_n(A) \mathbf{v}$ efficiently using the recursive relation of Chebyshev polynomials, that is,

$$T_{j+1}(x) = 2x T_j(x) - T_{j-1}(x),$$

where $T_j(x)$ is the Chebyshev polynomial of the first-kind with degree $j$. Let $\mathbf{w}_j := T_j(A) \mathbf{v}$ for $j \geq 0$, and we have that

$$\frac{\partial}{\partial \theta_i} \mathbf{v}^\top p_n(A) \mathbf{v} = \frac{\partial}{\partial \theta_i} \left( \sum_{j=0}^n b_j \mathbf{v}^\top T_j(A) \mathbf{v} \right) = \sum_{j=0}^n b_j \mathbf{v}^\top \left( \frac{\partial}{\partial \theta_i} T_j(A) \mathbf{v} \right) = \sum_{j=0}^n b_j \mathbf{v}^\top \frac{\partial \mathbf{w}_j}{\partial \theta_i}. \quad (32)$$

In the right hand side, $\mathbf{v}^\top \left( \frac{\partial \mathbf{w}_j}{\partial \theta_i} \right)$ can be computed using the recursion $\mathbf{w}_{j+1} = 2A \mathbf{w}_j - \mathbf{w}_{j-1}$:

$$\mathbf{v}^\top \frac{\partial \mathbf{w}_{j+1}}{\partial \theta_i} = \mathbf{v}^\top \frac{\partial}{\partial \theta_i} (2A \mathbf{w}_j - \mathbf{w}_{j-1}) = 2 \, \mathbf{v}^\top \frac{\partial}{\partial \theta_i} (A \mathbf{w}_j) - \mathbf{v}^\top \frac{\partial \mathbf{w}_{j-1}}{\partial \theta_i}$$

$$= 2 \left( \mathbf{v}^\top \frac{\partial A}{\partial \theta_i} \mathbf{w}_j + \mathbf{v}^\top A \frac{\partial \mathbf{w}_j}{\partial \theta_i} \right) - \mathbf{v}^\top \frac{\partial \mathbf{w}_{j-1}}{\partial \theta_i}$$

where $\mathbf{v}^\top \frac{\partial \mathbf{w}_1}{\partial \theta_i} = \mathbf{v}^\top \frac{\partial A}{\partial \theta_i} \mathbf{v}$ and $\mathbf{v}^\top \frac{\partial \mathbf{w}_0}{\partial \theta_i} = 0$. Applying induction on $j \geq 1$, we can obtain that

$$\mathbf{v}^\top \frac{\partial \mathbf{w}_{j+1}}{\partial \theta_i} = \sum_{k=0}^j (2 - \mathbb{1}_{k=0}) \, \mathbf{w}_k^\top \frac{\partial A}{\partial \theta_i} \mathbf{y}_{j-k}, \quad (33)$$

where $\mathbf{y}_{j+1} = 2A \mathbf{y}_j - \mathbf{y}_{j-1} = 2 \mathbf{w}_{j+1} + \mathbf{y}_{j-1}, \mathbf{y}_1 = 2A \mathbf{v}$ and $\mathbf{y}_0 = \mathbf{v}$. [3] Putting (33) to (32), we get

$$\frac{\partial}{\partial \theta_i} \mathbf{v}^\top p_n(A) \mathbf{v} = \sum_{j=0}^{n-1} b_{j+1} \mathbf{v}^\top \frac{\partial \mathbf{w}_{j+1}}{\partial \theta_i} = \sum_{j=0}^{n-1} b_{j+1} \left( \sum_{k=0}^j (2 - \mathbb{1}_{k=0}) \mathbf{w}_k^\top \frac{\partial A}{\partial \theta_i} \mathbf{y}_{j-k} \right). \quad (34)$$

In case when $A = \theta \theta^\top + \varepsilon I$ and $\theta \in \mathbb{R}^{d \times r}$, it holds that for $\ell = 1, \ldots, d$ and $m = 1, \ldots, r$,

$$\frac{\partial A}{\partial \theta_{\ell,m}} = \mathbf{e}_\ell \theta_{:,m}^\top + \theta_{:,m} \mathbf{e}_\ell^\top, \quad (35)$$

where $\theta_{:,m} \in \mathbb{R}^d$ is the $m$-th column of $\theta$ and $\mathbf{e}_\ell \in \mathbb{R}^d$ is a unit vector with the index $\ell$. Finally, we substitute (35) to (34) to have

$$
\begin{aligned}
\left[\frac{\partial}{\partial\theta}\mathbf{v}^\top p_n(A)\mathbf{v}\right]_{\ell,m} &= \frac{\partial}{\partial\theta_{\ell,m}}\mathbf{v}^\top p_n(A)\mathbf{v} = \sum_{j=0}^{n-1} b_{j+1}\left(\sum_{k=0}^{j}(2-\mathbb{1}_{k=0})\,\mathbf{w}_k^\top\left(\frac{\partial A}{\partial\theta_{\ell,m}}\right)\mathbf{y}_{j-k}\right) \\
&= \sum_{j=0}^{n-1} b_{j+1}\left(\sum_{k=0}^{j}(2-\mathbb{1}_{k=0})\,\mathbf{w}_k^\top\left(\mathbf{e}_\ell\theta_{:,m}^\top + \theta_{:,m}\mathbf{e}_\ell^\top\right)\mathbf{y}_{j-k}\right) \\
&\overset{(\dagger)}{=} \sum_{j=0}^{n-1} b_{j+1}\left(\sum_{k=0}^{j}(2-\mathbb{1}_{k=0})\left(\mathbf{e}_\ell^\top\mathbf{w}_k\mathbf{y}_{j-k}^\top\theta_{:,m} + \mathbf{e}_\ell^\top\mathbf{y}_{j-k}\mathbf{w}_k^\top\theta_{:,m}\right)\right) \\
&= \mathbf{e}_\ell^\top\left[\sum_{j=0}^{n-1} b_{j+1}\left(\sum_{k=0}^{j}(2-\mathbb{1}_{k=0})\left(\mathbf{w}_k\mathbf{y}_{j-k}^\top + \mathbf{y}_{j-k}\mathbf{w}_k^\top\right)\right)\right]\theta_{:,m} \\
&\overset{(\ddagger)}{=} \mathbf{e}_\ell^\top\left[\sum_{j=0}^{n-1} b_{j+1}\left(\sum_{k=0}^{j}(2-\mathbb{1}_{k=0})\,2\,\mathbf{w}_k\mathbf{y}_{j-k}^\top\right)\right]\theta_{:,m} \\
&= \mathbf{e}_\ell^\top\left[2\sum_{j=0}^{n-1} b_{j+1}\left(\sum_{k=0}^{j}(2-\mathbb{1}_{k=0})\,\mathbf{w}_k\mathbf{y}_{j-k}^\top\right)\theta\right]\mathbf{e}_m' \\
&= \left[2\sum_{k=0}^{n-1}(2-\mathbb{1}_{k=0})\,\mathbf{w}_k\left(\sum_{j=k}^{n-1} b_{j+1}\mathbf{y}_{j-k}\right)^\top\theta\right]_{\ell,m}
\end{aligned}
$$

where $\mathbf{e}_m' \in \mathbb{R}^r$ is the unit vector with index $m$ satisfying with $\theta_{:,m} = \theta\mathbf{e}_m'$. The equality $(\dagger)$ holds from that $\mathbf{a}^\top\mathbf{b} = \mathbf{b}^\top\mathbf{a}$ for any two vectors $\mathbf{a}$ and $\mathbf{b}$, and for the equality $(\ddagger)$ it is easy to check that $\sum_{k=0}^{j}(2-\mathbb{1}_{k=0})\,\mathbf{w}_k\mathbf{y}_{j-k}^\top = \sum_{k=0}^{j}(2-\mathbb{1}_{k=0})\,\mathbf{y}_{j-k}\mathbf{w}_k^\top$ using $2\mathbf{w}_j = \mathbf{y}_j - \mathbf{y}_{j-2}$ for $j \geq 2$. Thus,

$$
\nabla_\theta\mathbf{v}^\top p_n(A)\mathbf{v} = 2\sum_{k=0}^{n-1}(2-\mathbb{1}_{k=0})\,\mathbf{w}_k\left(\sum_{j=k}^{n-1} b_{j+1}\mathbf{y}_{j-k}\right)^\top\theta
$$

This completes the proof of Lemma 7.

### B.5.5 Proof of Lemma 8

The proof of Lemma 8 is similar with the proof of Lemma 4. We recall the formulation

$$
\psi_i := \frac{\partial}{\partial\theta_i}\mathbf{v}^\top\widehat{p}_n\left(A(\theta)\right)\mathbf{v} = \frac{2}{b-a}\mathbf{v}^\top G\mathbf{v}
$$

where

$$
G = \sum_{j=0}^{n-1}\widehat{b}_{j+1}\left(2\sum_{r=0}^{j}{}'T_r(\widetilde{A})\frac{\partial A}{\partial\theta_i}U_{j-r}(\widetilde{A})\right)
$$

and $\widetilde{A} = \frac{2}{b-a}A(\theta) - \frac{b+a}{b-a}I$. Define that $\Delta G := G(\theta) - G(\theta')$. Our goal is to find some $\beta \in \mathbb{R}$ such that $\mathbf{E}_{n,\mathbf{v}}[(\mathbf{v}^\top\Delta G\mathbf{v})^2] \leq \beta^2(\theta_i - \theta_i')^2$. For notational simplicity, we write that

$$
\Delta T_r := T_r(\widetilde{A}) - T_r(\widetilde{A}') = T_r - T_r', \qquad \Delta U_j := U_j(\widetilde{A}) - U_j(\widetilde{A}') = U_j - U_j',
$$

$$
\Delta A := \frac{2}{b-a}\left(A(\theta) - A(\theta')\right), \quad \Delta\frac{\partial A}{\partial\theta} := \frac{\partial A(\theta)}{\partial\theta} - \frac{\partial A(\theta')}{\partial\theta}, \quad \Delta\theta = \theta - \theta'.
$$

and $\Delta G$ can be expressed as

$$
\Delta G = \sum_{j=0}^{n-1}\widehat{b}_{j+1}\left(2\sum_{r=0}^{j}{}'T_r\frac{\partial A}{\partial\theta_i}U_{j-r} - T_r'\frac{\partial A'}{\partial\theta_i}U_{j-r}'\right).
$$

We use similar procedure in the proof of Lemma 4 to obtain

$$
\frac{(b-a)^2}{4}\mathbf{E}_{n,\mathbf{v}}\left[(\psi_i-\psi_i')^2\right]=\mathbf{E}_{n,\mathbf{v}}\left[\left(\mathbf{v}^\top\Delta G\mathbf{v}\right)^2\right]=\mathbf{E}_n\left[\mathbf{E}_{\mathbf{v}}\left[\left(\mathbf{v}^\top\Delta G\mathbf{v}\right)^2\big|n\right]\right]
$$

$$
=\mathbf{E}_n\left[\mathrm{Var}_{\mathbf{v}}[\mathbf{v}^\top\Delta G\mathbf{v}|n]+\mathbf{E}_{\mathbf{v}}\left[\mathbf{v}^\top\Delta G\mathbf{v}|n\right]^2\right]
$$

$$
\leq\mathbf{E}_n\left[2\left\|\Delta G\right\|_F^2+\left(\mathrm{tr}\left(\Delta G\right)\right)^2\right]. \tag{36}
$$

For the first term in (36), we use the triangle inequality to obtain

$$
\left\|\Delta G\right\|_F\leq\sum_{j=0}^{n-1}\left|\widehat{b}_{j+1}\right|\left(2\underbrace{\sideset{}{'}\sum_{r=0}^{j}\left\|T_r\frac{\partial A}{\partial\theta_i}U_{j-r}-T_r'\frac{\partial A'}{\partial\theta_i}U_{j-r}'\right\|_F}_{(\ddagger)}\right)
$$

and consider that

$$
(\ddagger)\leq\left\|(T_r-T_r')\frac{\partial A}{\partial\theta_i}U_{j-r}\right\|_F+\left\|T_r\frac{\partial A}{\partial\theta_i}\left(U_{j-r}-U_{j-r}'\right)\right\|_F+\left\|T_r'\left(\frac{\partial A}{\partial\theta_i}-\frac{\partial A'}{\partial\theta_i}\right)U_{j-r}'\right\|_F
$$

$$
\leq\left\|\Delta T_r\right\|_2\left\|\frac{\partial A}{\partial\theta_i}\right\|_F\left\|U_{j-r}\right\|_2+\left\|T_r\right\|_2\left\|\frac{\partial A}{\partial\theta_i}\right\|_F\left\|\Delta U_{j-r}\right\|_2+\left\|T_r'\right\|_2\left\|\Delta\frac{\partial A}{\partial\theta_i}\right\|_F\left\|U_{j-r}\right\|_2
$$

$$
\leq\left\|\Delta A\right\|_2 r^2\left\|\frac{\partial A}{\partial\theta_i}\right\|_F(j-r+1)+\left\|\frac{\partial A}{\partial\theta_i}\right\|_F\frac{(j-r)(j-r+1)(j-r+2)}{3}\left\|\Delta A\right\|_2+\left\|\Delta\frac{\partial A}{\partial\theta_i}\right\|_F(j-r+1)
$$

where the first inequality is from the triangle inequality of $\left\|\cdot\right\|_F$ and the second inequality holds from $\left\|XY\right\|_F\leq\left\|X\right\|_2\left\|Y\right\|_F$ for multiplicable matrices $X,Y$ and the last is from $\left\|T_i(\widetilde{A})\right\|_2\leq 1$, $\left\|U_i(\widetilde{A})\right\|_2\leq i+1$ for $i\geq 0$ and

$$
\left\|U_i(X+E)-U_i(X)\right\|_2\leq\frac{i(i+1)(i+2)}{3}\left\|E\right\|_2 \tag{37}
$$

for $X,E\in\mathcal{S}^{d\times d}$ satisfying with $\left\|X+E\right\|_2,\left\|X\right\|_2\leq 1$ (see Section B.5.8).
Summing $(\ddagger)$ for all $r=0,1,\ldots,j$, we have

$$
\left\|\Delta G\right\|_F\leq\sum_{j=0}^{n-1}\left|\widehat{b}_{j+1}\right|\left(\left\|\Delta A\right\|_2\left\|\frac{\partial A}{\partial\theta_i}\right\|_F\frac{j(j+1)^2(j+2)}{3}+\left\|\Delta\frac{\partial A}{\partial\theta_i}\right\|_F(j+1)^2\right)
$$

$$
\leq\max\left(\left\|\Delta A\right\|_2\left\|\frac{\partial A}{\partial\theta_i}\right\|_F,\left\|\Delta\frac{\partial A}{\partial\theta_i}\right\|_F\right)\sum_{j=0}^{n-1}\left|\widehat{b}_{j+1}\right|\left(\frac{j(j+1)^2(j+2)}{3}+(j+1)^2\right)
$$

$$
\leq\frac{1}{2}\max\left(\left\|\Delta A\right\|_2\left\|\frac{\partial A}{\partial\theta_i}\right\|_F,\left\|\Delta\frac{\partial A}{\partial\theta_i}\right\|_F\right)\sum_{j=0}^{n-1}\left|\widehat{b}_{j+1}\right|(j+1)^4.
$$

If one estimates $\psi$ and $\psi'$ using $M$ Rademacher random vectors, the variance of $\mathbf{v}^\top\Delta G\mathbf{v}$ is reduced by $1/M$ so that we have

$$
2\left\|\Delta G\right\|_F^2\leq\frac{1}{2M}\max\left(\left\|\Delta A\right\|_2^2\left\|\frac{\partial A}{\partial\theta_i}\right\|_F^2,\left\|\Delta\frac{\partial A}{\partial\theta_i}\right\|_F^2\right)\left(\sum_{j=1}^{n}\left|\widehat{b}_j\right|j^4\right)^2
$$

For the second term in (36), it holds that

$$\mathrm{tr}\left(\Delta G\right) = \mathrm{tr}\left(\frac{\partial A}{\partial \theta_i}\left(\widehat{p}'_n(A) - \widehat{p}'_n(A')\right)\right) \le \left\|\frac{\partial A}{\partial \theta_i}\right\|_F \left\|\widehat{p}'_n(A) - \widehat{p}'_n(A')\right\|_F$$

$$\le \left\|\frac{\partial A}{\partial \theta_i}\right\|_F \sum_{j=1}^n \left|\widehat{b}_j\right| j \left\|U_{j-1}(\widetilde{A}) - U_{j-1}(\widetilde{A}')\right\|_F$$

$$\le \left\|\frac{\partial A}{\partial \theta_i}\right\|_F \|\Delta A\|_F \sum_{j=1}^n \left|\widehat{b}_j\right| \frac{(j^2-1)j^2}{3}$$

$$\le \left\|\frac{\partial A}{\partial \theta_i}\right\|_F \frac{\|\Delta A\|_F}{3} \sum_{j=1}^n \left|\widehat{b}_j\right| j^4.$$

where the inequality in the first line holds from matrix version Cauchy-Schwarz inequality, the inequality in the second line holds from $\widehat{p}'_n(x) = \left(\sum_{j=0}^n \widehat{b}_j T_j(x)\right)' = \sum_{j=1}^n \widehat{b}_j j U_{j-1}(x)$ and inequality in the third line holds from (37).

Putting all together into (36), we obtain that

$$\mathbf{E}_{n,\mathbf{v}}\left[(\psi_i - \psi'_i)^2\right] = \mathbf{E}_n\left[2\|\Delta G\|_F^2 + (\mathrm{tr}\left(\Delta G\right))^2\right]$$

$$\le \left(\frac{1}{2M}\max\left(\|\Delta A\|_2^2 \left\|\frac{\partial A}{\partial \theta_i}\right\|_F^2, \left\|\Delta\frac{\partial A}{\partial \theta_i}\right\|_F^2\right) + \left\|\frac{\partial A}{\partial \theta_i}\right\|_F^2 \frac{\|\Delta A\|_F^2}{9}\right)\mathbf{E}_n\left[\left(\sum_{j=1}^n \left|\widehat{b}_j\right| j^4\right)^2\right]$$

$$\le \left(\left(\frac{1}{2M} + \frac{1}{9}\right)\left\|\frac{\partial A}{\partial \theta_i}\right\|_F^2 \|\Delta A\|_F^2 + \frac{1}{2M}\left\|\Delta\frac{\partial A}{\partial \theta_i}\right\|_F^2\right)\mathbf{E}_n\left[\left(\sum_{j=1}^n \left|\widehat{b}_j\right| j^4\right)^2\right]$$

$$\le \left(\left(\frac{1}{2M} + \frac{1}{9}\right)\left\|\frac{\partial A}{\partial \theta_i}\right\|_F^2 \frac{4L_A^2\|\Delta\theta\|_2^2}{(b-a)^2} + \frac{1}{2M}\left\|\Delta\frac{\partial A}{\partial \theta_i}\right\|_F^2\right)\mathbf{E}_n\left[\left(\sum_{j=1}^n \left|\widehat{b}_j\right| j^4\right)^2\right]$$

where the inequality in the second line holds from $\max(a,b) \le a+b$ for $a,b \in \mathbb{R}^+$ and the inequality in the third line holds from the Lipschitz continuity on $A$ (assumption $\mathcal{A}(2)$), formally,

$$\|A(\theta) - A(\theta')\|_2 \le \|A(\theta) - A(\theta')\|_F \le L_A \|\theta - \theta'\|_2.$$

Summing the above for all $i = 1, 2, \ldots, d'$ and using that $\|\partial A/\partial\theta\|_F \le L_A$ and $\|\Delta(\partial A/\partial\theta)\|_F \le \beta_A\|\Delta\theta\|_2$, we get

$$\mathbf{E}_{n,\mathbf{v}}\left[\|\psi - \psi'\|_2^2\right] \le D_0\left(\frac{L_A^4 + \beta_A^2}{M} + L_A^4\right)\|\Delta\theta\|_2^2 \mathbf{E}_n\left[\left(\sum_{j=1}^n \left|\widehat{b}_j\right| j^4\right)^2\right]$$

for some constant $D_0 > 0$.

To bound the right-hand side, we introduce the following lemma, whose proof is in Section B.5.7.

**Lemma 10** *Suppose that $q_n^*$ is the optimal degree distribution as defined in (11) and $b_j$ is the Chebyshev coefficients of analytic function $f$. Define the weighted coefficient $\widehat{b}_j$ as $\widehat{b}_j = b_j/(1 - \sum_{i=0}^{j-1} q_i^*)$ for $j \ge 0$ and conventionally $q_{-1}^* = 0$. Then, there exists constants $D_1', D_2' > 0$ independent of $M, N$ such that*

$$\sum_{n=1}^\infty q_n^*\left(\sum_{j=1}^n |\widehat{b}_j| j^4\right)^2 \le D_1' + \frac{D_2' N^8}{\rho^{2N}}.$$

Therefore, we obtain the result that

$$\mathbf{E}_{n,\mathbf{v}} \left[ \|\psi - \psi'\|_2^2 \right] \leq \beta^2 \|\theta - \theta'\|_2^2 \tag{38}$$

where

$$\beta^2 := \left( \frac{L_A^4 + \beta_A^2}{M} + L_A^4 \right) \left( D_1 + \frac{D_2 N^8}{\rho^{2N}} \right)$$

Under the assumption that $g(\theta)$ is $\beta_g$-smooth function (assumptio $\mathcal{A}(2)$,), we have that

$$\|\nabla g(\theta) - \nabla g(\theta')\|_2^2 \leq \beta_g^2 \|\theta - \theta'\|_2^2. \tag{39}$$

Summing both (38) and (39), it yields that

$$\mathbf{E}_{n,\mathbf{v}} \left[ \|\psi - \psi'\|_2^2 + \|\nabla g(\theta) - \nabla g(\theta')\|_2^2 \right] \leq \left( \beta^2 + \beta_g^2 \right) \|\theta - \theta'\|_2^2.$$

Using $\|a + b\| \leq 2(\|a\|^2 + \|b\|^2)$ again, we conclude that

$$\mathbf{E}_{n,\mathbf{v}} \left[ \|\psi + \nabla g(\theta) - \psi' - \nabla g(\theta')\|_2^2 \right] \leq 2 \left( \beta^2 + \beta_g^2 \right) \|\theta - \theta'\|_2^2.$$

This completes the proof of Lemma 8.

### B.5.6 Proof of Lemma 9

Recall that the optimal degree distribution as

$$q_i^* = \begin{cases} 0 & \text{for } i < K \\ 1 - (N - K)\,(\rho - 1)\rho^{-1} & \text{for } i = K \\ (N - K)(\rho - 1)^2 \rho^{-i-1+K} & \text{for } i > K. \end{cases}$$

where $K = \max(0, N - \lfloor \frac{\rho}{\rho-1} \rfloor)$. We first use the upper bound on the coefficients from (2), i.e., $|b_j| \leq 2U/\rho^j$ to obtain

$$\sum_{n=1}^{\infty} q_n^* \left( \sum_{j=1}^{n} |\widehat{b}_j| j^2 \right)^2 = \sum_{n=K}^{\infty} q_n^* \left( \sum_{j=1}^{n} |\widehat{b}_j| j^2 \right)^2 \leq 4U^2 \sum_{n=K}^{\infty} q_n^* \left( \sum_{j=1}^{n} \frac{j^2}{(1 - \sum_{i=0}^{j-1} q_i^*)\rho^j} \right)^2 \tag{40}$$

To express (40) more simple, we define that

$$\Lambda := \sum_{j=1}^{K} \frac{j^2}{(1 - \sum_{i=0}^{j-1} q_i^*)\rho^j} = \sum_{j=1}^{K} \frac{j^2}{\rho^j} \leq \frac{\rho(\rho + 1)}{(\rho - 1)^3}$$

which equals to the second term in the summation (40) when $n = K$. For $n \geq K + i, i \geq 1$, we get

$$\sum_{j=1}^{K+i} \frac{j^2}{(1 - \sum_{i=0}^{j-1} q_i^*)\rho^j} = \Lambda + \frac{\sum_{j=1}^{i}(K + j)^2}{(N - K)(\rho - 1)\rho^K}. \tag{41}$$

Putting $q_i^*$ and (41) to the right hand side of (40), we have

$$\left( 1 - (N - K)\frac{\rho - 1}{\rho} \right) \Lambda^2 + (N - K) \left( \frac{\rho - 1}{\rho} \right)^2 \left( \Lambda + \frac{(K + 1)^2}{(N - K)(\rho - 1)\rho^K} \right)^2$$

$$+ (N - K) \left( \frac{\rho - 1}{\rho} \right)^2 \frac{1}{\rho} \left( \Lambda + \frac{\sum_{j=1}^{2}(K + j)^2}{(N - K)(\rho - 1)\rho^K} \right)^2$$

$$+ (N - K) \left( \frac{\rho - 1}{\rho} \right)^2 \frac{1}{\rho^2} \left( \Lambda + \frac{\sum_{j=1}^{3}(K + j)^2}{(N - K)(\rho - 1)\rho^K} \right)^2$$

$$+ \cdots .$$

Rearranging all terms with respect to $\Lambda$, we obtain that

$$\Lambda^2 + \frac{2(\rho-1)}{\rho^{K+1}}\left(\sum_{i=1}^{\infty}\frac{\sum_{j=1}^{i}(K+j)^2}{\rho^i}\right)\Lambda + \frac{1}{(N-K)\rho^{2K+1}}\left(\sum_{i=1}^{\infty}\frac{\left(\sum_{j=1}^{i}(K+j)^2\right)^2}{\rho^i}\right).$$

Note that

$$\sum_{i=1}^{\infty}\frac{\sum_{j=1}^{i}(K+j)^2}{\rho^i} = \frac{K^2\rho(\rho-1)^2 + 2K\rho^2(\rho-1) + \rho^2(\rho+1)}{(\rho-1)^4}$$

and

$$\sum_{i=1}^{\infty}\frac{(\sum_{j=1}^{i}(K+j)^2)^2}{\rho^i} = \texttt{poly}(K^4).$$

Since $K = O(N)$ and $N - K = O(1)$, we can conclude that

$$\sum_{n=1}^{\infty}q_n^*\left(\sum_{j=1}^{n}|\widehat{b}_j|j^2\right)^2 \leq C_1 + C_2\frac{N^4}{\rho^{2N}}$$

for some constants $C_1, C_2 > 0$ not depend on $N$.

### B.5.7    Proof of Lemma 10

The proof of Lemma 10 is straightforward from that of Lemma 9. One can replace $j^2$ into $j^4$ in the proof of Lemma 9, which results in $N^8$ dependence. We omit the details of the proof.

### B.5.8    Proof of other lemmas

**Lemma 11** *Suppose that $A, A + E \in \mathbb{R}^{d\times d}$ are symmetric matrices and they have eigenvalues in $[-1, 1]$. Let $T_i$ and $U_i$ be the first and the second kind of Chebyshev basis polynomial with degree $i \geq 0$, respectively. Then, it holds that*

$$\|T_i(A + E) - T_i(A)\| \leq i^2\|E\|, \quad \|U_i(A + E) - U_i(A)\| \leq \frac{i(i+1)(i+2)}{3}\|E\|.$$

*where $\|\cdot\|$ can be $\|\cdot\|_2$ (spectral norm) or $\|\cdot\|_F$ (Frobenius norm).*

**Proof.** Denote $R_i := T_i(A + E) - T_i(A)$. From the recursive relation of Chebyshev polynomial, i.e., $T_{j+1}(x) = 2AT_j(x) - T_{j-1}(x)$, $R_i$ has following property:

$$R_{i+1} = 2(A + E)R_i - R_{i-1} + 2E\,T_i(A)$$

for $i \geq 1$ where $R_1 = E$, $R_0 = \mathbf{0}$. By induction on $i$, it is easy to show that

$$R_{i+1} = 2\sum_{j=0}^{i}{}'U_{i-j}(A + E)\,E\,T_j(A)$$

where $U_j(x)$ is the Chebyshev polynomial of the second kind. Therefore, we have

$$\|R_{i+1}\|_F \leq 2\sum_{j=0}^{i}{}'\|U_{i-j}(A + E)\,E\,T_j(A)\|_F$$

$$\leq 2\sum_{j=0}^{i}{}'\|U_{i-j}(A + E)\|_2\,\|E\|_F\,\|T_j(A)\|_2$$

$$\leq 2\sum_{j=0}^{i}{}'(i + 1 - j)\,\|E\|_F = (i + 1)^2\,\|E\|_F$$

where the second inequality holds from $\|YX\|_F = \|XY\|_F \le \|X\|_2 \|Y\|_F$ for matrices $X, Y$. This also holds for $\|\cdot\|_2$ giving that $\|R_{i+1}\|_2 \le (i+1)^2 \|E\|_2$. Similarly, we denote $Y_i := U_i(A+E) - U_i(A)$. By induction on $i$, it is easy to show that

$$Y_{i+1} = 2 \sum_{j=0}^{i} U_{i-j}(A+E) E U_j(A)$$

Then, we have that for $i \ge 0$

$$\|Y_{i+1}\|_F \le 2 \sum_{j=0}^{i} \|U_{i-j}(A+E) E U_j(A)\|_F$$

$$\le 2 \sum_{j=0}^{i} \|U_{i-j}(A+E)\|_2 \|E\|_F \|U_j(A)\|_2$$

$$\le 2 \sum_{j=0}^{i} (i+1-j)(j+1) \|E\|_F$$

$$= \frac{(i+1)(i+2)(i+3)}{3} \|E\|_F .$$

This also holds for $\|\cdot\|_2$ giving that $\|Y_{i+1}\|_2 \le \frac{(i+1)(i+2)(i+3)}{3} \|E\|_2$. This completes the proof of Lemma 11. ∎

**Lemma 12** *For symmetric matrices $A, B \in \mathcal{S}^{d \times d}$, it holds that $\mathtt{tr}(AB) \le \|A\|_{\mathrm{nuc}} \|B\|_2$.*

**Proof.** Since $A$ is real symmetric, it can be written as $A = \sum_{i=1}^{d} \lambda_i \mathbf{u}_i \mathbf{u}_i^\top$ where $\lambda_i$ and $\mathbf{u}_i$ is $i$-th eigenvalue and eigenvector, respectively. Then, the result follows that

$$\mathtt{tr}(AB) = \sum_{i=1}^{d} \lambda_i \, \mathtt{tr}\left(\mathbf{u}_i \mathbf{u}_i^\top B\right) = \sum_{i=1}^{d} \lambda_i \, \mathbf{u}_i^\top B \mathbf{u}_i$$

$$\le \sum_{i=1}^{d} |\lambda_i| \, \mathbf{u}_i^\top B \mathbf{u}_i$$

$$\le \sum_{i=1}^{d} |\lambda_i| \|B\|_2 = \|A\|_{\mathrm{nuc}} \|B\|_2.$$

This completes the proof of Lemma 12. ∎