[Reviews · NeurIPS 2018]

Reviewer 1



Spectral optimization is defined as finding $\theta$ that minimizes $F(A(\theta)) + g(\theta)$ where $A(\theta)$ is a symmetric matrix and $F$ typically the trace of an analytic function i.e. $F(A) = tr(p(A))$ where $p$ is a polynomial. They propose an unbiased estimator of $F$ by randomly truncating the Chebyshev approximation to $F$ and doing importance sampling. Moreover, they calculate the optimal distribution for this importance sampling. They demonstrate how this method would be used for SGD and stochastic Variance Reduced Gradient. Finally, they demonstrate two applications---matrix completion and minimizing log determinant for fitting Gaussian processes. This paper is very clearly written, given the theoretical depth of the work. The main result is the optimal distribution for random truncation. The authors do an excellent job of defining and motivating this problem, proving their result, and demonstrating its effectiveness in two applications. I would strongly argue for accepting this paper. ## Questions ## 1. As pointed out in the paper, several other papers have taken similar approaches to approximating $tr(f(A))$ using Chebyshev approximations e.g. [6, 16, 28, 25]. However, I haven't seen a proof that importance sampling truncated series is strictly better than importance sampling individual terms. Have you been able to prove such a result? 2. In algorithm 1 and 2, it seems like the only stochasticity comes from sampling Rademacher vectors. We can imagine a case where $A(\theta)$ can only be calculated relative to a specific batch of data (e.g. the Hessian of a NN for a specific batch). Are you specifically not treating this case, or do you believe you have accounted for it here? 3. To make the two applications more explicit for people not very familiar with GPs or matrix completion, I would suggest writing out the objective that is being minimized in a form similar to $F(A(\theta)) + g(\theta)$ just to make the correspondence explicit.

Reviewer 2



The paper considers stochastic Chebyshev gradient descent for spectral optimization. Specifically, the paper develops unbiased stochastic gradients by combining randomized trace estimator with stochastic truncation of the Chebyshev expansion. This unbiased stochastic gradient is used in SGD and SVRG, and the experimental results on two applications demonstrate the effectiveness. This Chebyshev expansion is based on a bound interval, and has two parameters a and b. Is this a limitation? For applications without accurate estimation on a and b, how can one construct the estimator? There are quite some other types of methods in the literatures for solving the matrix completion problem. The results will be more convincing, if there are some comparison with them. The proposed SGD and SVRG frameworks mainly focus on the unconstrained problems. Can they also be applied on formulations with complex constraints?

Reviewer 3



This work aims to proposed a stochastic gradient descent for spectral optimization. Unbiased stochastic gradients for spectral-sums are proposed based on the Chebyshev expansions. Besides the calculation of stochastic gradients, the truncation distribution is also designed for fast and stable convergence. It is worth noting that the major difficulties of applying Chebyshev approximations to spectral-sums has also been addressed in [1]. Here are the comments: To calculate the gradients, one need to draw M Rademacher vectors. What the value of M is used in the experiments? Is M sensitive to the results? How to determine this parameter? Followed by the above comment, memory requirement is also a target of interest in stochastics learning, it is suggested to include the analysis of space complexity as well. It is also encouraged to apply the proposed method to the rank minimization problems, such as the robust PCA [2] or robust subspace discovery [3], where the nuclear norm is used. Although the authors mentioned that Chebyshev approximations are nearly optimal in approximation among polynomial series theoretically, it is suggested that a comparison with standard SGD (i.e., Taylor expansions) Please specify the notation ψ^((t))and ψ ̃^((s)) in Algorithm 1 and 2 [1] Han, Insu, Malioutov, Dmitry, Avron, Haim, and Shin, Jinwoo. Approximating spectral sums of large-scale matrices using stochastic chebyshev approximations. SIAM Journal on Scientific Computing, 39(4):A1558–A1585, 2017. [2] Xinggang Wang, Zhengdong Zhang, Yi Ma, Xiang Bai, Wenyu Liu, and Zhuowen Tu. “Robust Subspace Discovery via Relaxed Rank Minimization”, Neural Computation, 26(3): 611-635, March 2014. [3] Emmanuel Candes, Xiaodong Li, Yi Ma, and John Wright. “Robust Principal Component Analysis”, vol. 58, issue 3, article 11, Journal of the ACM, May 2011.